# Bathymetry and latitude modify lake warming under ice

Cintia L. Ramón[1]*, Hugo N. Ulloa[2], Tomy Doda[1], Kraig B. Winters[3], Damien Bouffard[1]

[1]Department of Surface Waters – Research and Management, Eawag (Swiss Federal Institute of Aquatic Science and Technology), Kastanienbaum, 6047, Switzerland.
[2]Physics of Aquatic Systems Laboratory, EPFL (École Polytechnique Fédérale de Lausanne), Lausanne, CH-1015, Switzerland.
[3]Scripps Institution of Oceanography, University of California, San Diego, La Jolla, CA 92093-0209, USA

*Correspondence to*: Cintia L. Ramon (cintia.ramoncasanas@eawag.ch)

**Abstract.** In late winter, solar radiation is the main driver of water motion in ice-covered lakes. The resulting circulation and mixing determine the spatial distribution of heat within the lake and affect the heat budget of the ice cover. Although under-ice lake warming is often modeled as a one-dimensional vertical process, lake bathymetry induces a relative excess heating of shallow waters, creating horizontal density gradients. This study shows that the dynamic response to these gradients depends sensitively on lake size and latitude —Earth rotation— and is controlled by the Rossby number. In the ageostrophic limit, horizontal density gradients drive cross-shore circulation that transports excess heat to the lake interior, accelerating the under-ice warming there. In the geostrophic regime, the circulation of the near- and off-shore waters decouple and excess heat is retained in the shallows. The flow regime controls the fate of this excess heat and its contribution to water-induced ice-melt.

## 1 Introduction

Convection in rotating fluids is a ubiquitous process redistributing heat in planetary systems (Afanasyev and Zhang, 2018; Heimpel et al., 2016). In the case of the Earth, its spin controls the outer core convective flows responsible for the 'geodynamo' (e.g. Buffett, 2000), as well as the thermohaline oceanic circulation, and the large-scale atmospheric 'cell' circulation (Vallis, 2017) and its climate (e.g. Cabré et al., 2017). Yet, as we move from planetary to smaller-scale systems, the importance of rotation in affecting convective processes has drawn, comparatively, less attention. This is the case in lakes. The classical example of convection in lakes is the winter cooling that controls deep mixing (Schwefel et al., 2016), but field observations have been able to characterize other convective processes (e.g., Bouffard and Wüest, 2019). For example, the presence of sloping lateral boundaries sets up lateral convective flows that connect the usually shallow littoral region with the deep lake interior (Fer et al., 2002; Monismith et al., 1990; Wells and Sherman, 2001). Shallow littoral waters heat or cool faster than the deeper offshore waters. This differential heating or cooling results in a cross-shore density gradient that triggers horizontal convection and downslope gravity currents. However, the influence of rotation on the distribution of heat is not well understood. The importance of Earth rotation on horizontal flows can be parameterized with the Rossby number, $Ro = U f^{-1} L^{-1}$. Here, $f$ is the Coriolis frequency, which increases with latitude, and $U$ and $L$ are the characteristic velocity and length

scales of the flow, respectively. Earth rotation affects horizontal flows when $Ro < 1$, and it is known, for example, to modify the pathways of rivers entering lakes (e.g., Davarpanah Jazi et al., 2020), and to decelerate the offshore progression of thermal bars (e.g., Holland et al., 2003) and to promote the formation of their associated basin-scale gyres (e.g., Huang, 1972). For the downslope gravity currents, the scales $U$ and $L$ correspond to the cross-shore velocity and the distance from the shore to the center of the lake basin, respectively. Gravity currents triggered by horizontal convection are usually "slow" ($U$ of O(cm s$^{-1}$), Fer et al., 2002; Monismith et al., 1990; Wells and Sherman, 2001), so their Rossby numbers are usually less than 1. Here we show that rotational effects on horizontal convection are especially relevant for ice-covered lakes in late winter since they modulate their warming regime.

Lake ice largely insulates the water column from the wind and from most of the atmospheric heat fluxes (Kirillin et al., 2012). Nevertheless, under thin snow cover conditions, solar radiation penetrates the ice and heats the water adjacent to it (Kirillin et al., 2012). In ice-covered freshwater lakes, water temperature is typically increasing from that of the freezing point at the ice-water interface, to the temperature of maximum water density, $T_{md}$ ($\sim 3.98°C$), near the bottom of the lake. The density of water can be estimated from its temperature through an equation of state. For temperature values below $T_{md}$, an increase in temperature results in an increase in density. Thus, by heating the uppermost coldest water, it becomes denser, causing it to plunge, initiating convectively-driven turbulence (Farmer, 1975). Radiatively-driven convection tends to both homogenize and heat the upper portion of the water column. The latter can have important implications for winter lake ecology (Farmer et al., 2015) and biogeochemical processes (e.g. Hampton et al., 2017; Karlsson et al., 2013). In addition, lake warming enhances ice melting by increasing the diffusive heat flux from the lake water to the ice. Having a robust mechanistic understanding of the processes controlling the vertical heat transport and distribution is especially relevant to assess the fate of lakes under climate change scenarios. Long term predictions based on one-dimensional (1D, vertical) models indicate that up to 25% of the current seasonally ice-covered lakes are projected to be permanently ice-free by the end of the 21st century (Woolway and Merchant, 2019). The latter might have broad impacts. However, such models and predictions do not integrate the effect of lateral boundaries in the vertical heat distribution.

Lake warming under ice is often modelled as a 1D vertical process (Farmer, 1975; Mironov et al., 2002). In the 1D framework, convective plumes driven by solar radiation lead to the formation of a convective mixed layer (CML, Fig.1a) above the relative quiescent stratified deep layers of the lake and below the diffusive layer at the ice-water interface (Farmer, 1975) (Fig. 1a). On a timescale of days to weeks during late winter, the cumulative insolation increases and the CML becomes warmer and deeper (e.g., Bouffard et al., 2019; Farmer, 1975), and, in shallow lakes, complete thermal mixing of the water column can occur before the end of the ice-on season (e.g., Huang et al., 2019; Salonen et al., 2014). However, even if the radiative flux through the ice remains spatially uniform over the lake area, the interaction of solar radiation and lake bathymetry can drive horizontal convection due to differential heating. Horizontal density gradients trigger gravity currents that propagate downslope and intrude into deep water at the base of the CML (Cortés and MacIntyre, 2020; Kirillin et al., 2015; Salonen et al., 2014; Stefanovic and Stefan, 2002) (Figs. 1b-c). Return flows towards the littoral regions develop at the top of the CML, leading to the formation of cross-shore circulation cells (Fig. 1b). Recently, it has been shown via 2D numerical experiments, that

horizontal advective fluxes of heat due to differential heating speed up the warming and deepening of the CML (Ulloa et al., 2019). The degree of departure from the 1D formulation increases as the fraction of the basin volume occupied by the littoral region —the region whose maximum depths are shallower than the CML depth— increases. By assuming that all the excess heat in the littoral region is effectively flushed into the lake interior, the underestimation of the contribution of advection and, thus, of the warming of the CML, can be quantified by a time dependent geometrical factor $G$ that accounts for a deepening

mixed layer and the bulk hypsometry of the basin (Ulloa et al., 2019)

$$G = \left| \frac{A_{shallow}}{A_{total}} \left( \frac{\bar{h}}{h_{cml}} - 1 \right) \right|, \tag{1}$$

where the vertical bars refer to absolute values, $A_{total}$ and $A_{shallow}$ are the surface areas of the lake and the littoral region (Fig. 1b and Fig. S1), respectively; and $\bar{h}$ and $h_{cml}$ are the average depth of the littoral region and that of the CML, respectively (Fig. 1d and details in Fig. S1). However, depending on the lake size — controlling the scale $L$ in $Ro$— and location (latitude) —

controlling $f$ in $Ro$ —, lake dynamics can be also strongly affected by Earth rotation, which affects the heat exchange between shallow and deep waters. There is field evidence on the influence of Coriolis in the circulation of lakes under ice (Forrest et al., 2013; Kirillin et al., 2015; Likens and Ragotzkie, 1966; Rizk et al., 2014) where the presence of horizontal cyclonic or anticyclonic gyres have been inferred or measured. These gyres can potentially modulate the advection of heat from the littoral region and modify lake warming.

Here we report, through 3D real-scale numerical simulations, that indeed for a fixed radiative heating, the resulting under-ice basin-scale heat distribution strongly depends on rotation, and, thus, potentially on the bathymetry and latitudinal location of a given lake. We found two distinctive dynamical regimes, controlled by the Rossby number, that result in a remarkably divergent distribution of incoming heat beneath the ice. In the ageostrophic regime, $Ro$ O($10^{-1}$-$10^0$), advection of heat from the littoral region effectively increases the deepening and warming rates of the CML in the lake interior. In the geostrophic

regime, $Ro \leq$ O($10^{-2}$), the dynamics of the littoral and the offshore region decouple: the excess heat is retained in the littoral region while the deepening and warming of the CML in the lake interior is well approximated using the 1D framework.

## 2. Methods

### 2.1 Scenarios

For validation and comparison purposes, our scenarios extend the simulations presented in Ulloa et al. (2019). We specifically

kept the same lake bathymetry as in their 2D spectral large eddy simulations (LES), but expanded into three dimensions to obtain a basin with radial symmetry (Fig. 1b). The fluid was initially at rest and linearly stratified, with $T = 0°C$ at the surface and $T = T_{md}$ at the maximum depth. We fix $T$ at 0°C at the ice-water boundary and subject the under-ice water to a laterally uniform but time and depth dependent radiative flux, $I(t,z)$ (°C m s$^{-1}$). This is equivalent to a surface heat flux, $Q(t,z) = c_p \rho_0 I(t,z)$ (W m$^{-2}$), with $c_p$ and $\rho_0$ being the heat capacity and reference density of fresh water near its freezing point, respectively.

The vertical distribution was modelled using Beer's law (Fig. 1a), whereas the time evolution was prescribed to roughly mimic the daily cycle of solar insolation (Fig. 1b and see Sect. 2.3). The forcing parameters reflect under-ice conditions in late winter. We investigate three scenarios ranging from weak ($Ro$ $O(10^{-1})$) to stronger ($Ro$ $O(10^{-3})$) rotational influence (Table 1 and see Sect. 2.4 and Sect. S1.1 in the supplementary material for the $Ro$ calculations). This range of $Ro$ spans the expected range of values typical of the varying size and latitudinal distribution of ice-covered lakes on Earth (see Sect. 4). We focus on a

rotational regime where convective-plume plunging and mixing occur on time scales much shorter than the inertial period ($\sim f^{-1}$), but the temperature gradients that result from differentially heating the shallow littoral and deep interior regions give rise to slower motions that are affected by rotation over longer time scales.

## 2.2 Computational model

Simulations were conducted with the RANS model MITgcm (MIT General Circulation Model, Marshall et al., 1997a, 1997b

and details in http://mitgcm.org). MITgcm is a three-dimensional, non-hydrostatic, z-coordinate, finite volume model that solves the Boussinesq form of the Navier-Stokes equations for incompressible fluids. The momentum equations are discretized on an Arakawa-C grid. The variables are advanced in time using a quasi-second-order Adams-Bashforth time-stepping scheme and preconditioned conjugate-gradient methods are used in the 2D and 3D inversion of the hydrostatic and non-hydrostatic pressure, respectively. The advection terms in the transport equation for temperature were discretized with the non-linear 3rd

order DST (direct space-time) with flux limiter. A modified nonlinear UNESCO equation of state (Jackett and Mcdougall, 1995) and the non-hydrostatic capabilities of MITgcm were used. Horizontal and vertical viscosities were parameterized with the isotropic 3D Smagorinsky approach (0.0005 coefficient). Background grid-dependent lateral viscosities were set to 0.002 (equivalent to a lateral eddy viscosity, $v_h$, of $\sim 8 \times 10^{-4}$ m$^2$ s$^{-1}$ for grid resolution in the x-direction, $\Delta x$, of 0.9 m and a time step, $\Delta t$, = 0.5 s). Background vertical viscosities, $v_z$, were set to $10^{-6}$ m$^2$ s$^{-1}$. Background horizontal and vertical diffusivities for

heat, $K_h$ and $K_z$, were set to $10^{-5}$ m$^2$ s$^{-1}$ and $1.4 \times 10^{-7}$ m$^2$ s$^{-1}$, respectively. No-slip conditions were applied at the lateral walls and the bottom. Though we could potentially have done this initial study using 3D LES, our interest extends to larger lakes over seasonal time scales and this setup provides and validates a starting point for such studies. MITgcm was originally built for ocean modelling, but has been applied satisfactorily to study lake hydrodynamics (e.g. Cimatoribus et al., 2018; Dorostkar et al., 2017). We further validate MITgcm against spectral LES (Ulloa et al., 2019) of radiatively heated ice-covered lakes (see

Sect. S1.5 in the supplementary material).

## 2.3 Under ice hydrodynamic and transport model

The lake bathymetry is conical and axisymmetric (Fig. 1), with a radius $R$ = 114 m and depths, $D$, increasing towards the lake interior in the radial direction, $r$, as in Ulloa et al. (2019):

$$D = H - \frac{H - d_{min}}{2}\left(1 - tanh\frac{r - r_0}{\beta}\right), \qquad (2)$$

where $H = 27.1$ m is the maximum height of the basin, $d_{min}$ (= 0.56 m) is the minimum depth in the littoral region, and $r_0$ (= 57.2 m) and $\beta$ (= 22.9 m) are topographic (slope) parameters. The physical domain was discretized using a horizontally uniform Cartesian grid ($\Delta x = \Delta y = 0.9$ m) with vertically variable thickness ($\Delta z$). $\Delta z$ increases with depth from $\Delta z = 0.05$ m within the first 2 m to a bottom cell of 0.3 m. In our simulations, temperature and pressure are the only scalars affecting density. The lake was initially quiescent with horizontal isotherms following a linear thermal stratification, with $T = 0$ ℃ at the ice-water

interface and $T = T_{md}$ at the bottom. Boundaries are prescribed as no slip and adiabatic, except at the ice-water interface. The ice-water boundary is modeled as a rigid lid with a constant temperature of 0 ℃. The radiative heat flux through the ice-water interface is specified as a sinusoidal daily cycle that attenuates with depth as in Ulloa et al. (2019):

$$I(t,z) = I_0 F(t) e^{-(H-z)/\lambda}. \tag{3}$$

Here the time $t$ is expressed in days, $I_0$ (= $1\times10^{-5}$ ℃ m s$^{-1}$) is the water surface radiative forcing, $F(t) = \sin(2\pi t)$ during the day

($t < 0.5$) and zero otherwise, and $1/\lambda$ (= 2.5 m$^{-1}$) is the depth scale for light attenuation. The order of magnitude of $I_0$ and the value for $\lambda$ selected, are representative of late-winter conditions in turbid waters (Bouffard et al., 2019; Leppäranta et al., 2003). The intensity of convection is expected to decrease as $\lambda$ increases (e.g., Winters et al., 2019), but the effect of light attenuation on differential heating and on the magnitude of the radial velocities remains secondary compared to the magnitude of $I_0$ and the geometry of the littoral region. For visualization purposes only, our results are shifted 0.25 days so the peak in

the radiative heat flux matches midday (Fig. 1).

**2.4 Rossby number and test cases**

Since we are interested in evaluating the advection of heat from the littoral to the lake interior, the surface radius, $R$, was selected as the characteristic length scale in the calculations of $Ro$. $Ro$ was calculated using the maximum radial velocity in the littoral region, $U_{rs\text{-}max}$, as the characteristic velocity scale (see details in Sect. S1.1 in supplementary material).

$$Ro = \frac{U_{rs-max}}{fR}. \tag{4}$$

$R/U_{rs\text{-}max}$ is the nominal time required for a gravity current at speed $U_{rs\text{-}max}$ to reach the center of the lake. When $R/U_{rs\text{-}max} > f^{-1}$, gravity currents are affected by Earth rotation (e.g., Davarpanah Jazi et al., 2020). While $R$ could vary several orders of magnitude among the ice-covered lakes on Earth, $f$ remains O($10^{-4}$) s$^{-1}$. Thus, $Ro$ depends predominantly on the horizontal dimensions of the lake. Still, the variability in $f$ alone allows that bigger lakes lie within the same rotation regime as one moves

towards lower latitudes. Note that for a given value of $U_{rs\text{-}max}$, to obtain the same value of $Ro$, Eq. (4) shows that a lake at 30° N ($f = 0.7\times10^{-4}$ s$^{-1}$) would still double the size of a lake at latitudes $\geq$ 70° N ($f \geq 1.4\times10^{-4}$ s$^{-1}$).

The model was first used to simulate rotational effects as in Ulloa et al. (2019), with a characteristic $Ro$ O($10^{-1}$). This corresponds to run 1 in Table1. For a range of measured radial velocities of O($10^{-3}$–$10^{-2}$) m s$^{-1}$ under ice (Forrest et al., 2013; Kirillin et al., 2015; Rizk et al., 2014), a value of $Ro$ O($10^{-1}$) could be representative of lakes of O($10^{-1}$-$10^0$) km in length (~2$L$).

To analyze the effect of rotation in the lake circulation and in the warming of the CML, two additional simulations were

conducted where we increased rotational effects by decreasing $Ro$ up to two orders of magnitude (runs 2 and 3 in Table 1). To analyze bathymetric effects (differential heating), an additional simulation was conducted (reference simulation) where forcing was kept as in run 1, but the bathymetry was modified to obtain a cylinder of depth $D = H$. Each run spans 12 radiative cycles (12 days). This number of cycles was long enough to expose and analyze the effect of rotation and bathymetry on lake warming under ice.

## 2.5 Differential heating, warming of the CML and heat balance

The magnitude of differential heating was quantified by comparing CML temperatures, $T_{cml}$, in the littoral region and at the lake center. We compared temperatures at 1 m depth (inside the CML in all runs and days > 1) at profile P1 in the littoral region (at a distance $\beta$ from the outer edge and ~91 m from the lake center) and at profile P2 at the lake center (Fig. 1c). The average $T_{cml}$ at the lake interior was calculated as the average mixed-layer temperature inside an interior control volume (Fig.1d) of radius ~25 m (as in Ulloa et al., 2019). This control volume was also used to calculate the importance of the advection of heat from the littoral region in the mixed-layer heat balance at the lake interior (see Sect. 3).

## 3. Results

### 3.1 Mixed layer warming and differential heating for different Rossby numbers

Radiatively-driven convection produces a CML with uniform temperature in each of the $Ro$ scenarios simulated (see the temperature profiles in Figs. 2a-c). This CML warms and deepens during the 12 daily cycles simulated, with higher temperatures reached as $Ro$ increases (red line in Figs. 2a-c). Fig. 2D shows the time history of temperature in the CML, $T_{cml}$, in the lake interior (average value in the interior control volume CV, Fig. 1d) for the three $Ro$ tested. The rate of warming of the interior mixing layer, $dT/dt$, increases with increasing $Ro$, with the net warming rate for $Ro$ $O(10^{-1})$ 23% higher on average than that for $Ro$ $O(10^{-3})$ (compare the red and blue lines in Fig. 2d).

The development and maintenance of a temperature contrast between the littoral and the lake interior is also affected by $Ro$. To quantify the differential heating, we compared the temperature difference at 1m depth (inside the CML), $\Delta T_{h,z=1m}$, between two profiles: one in the littoral region (P1, Fig. 1d) and one at the lake center (P2, Fig.1d). The temporal evolution of $\Delta T_{h,z=1m}$ is remarkably sensitive to $Ro$ (Fig. 2e). For $Ro$ $O(10^{-3})$ (blue line in Fig. 2e), $\Delta T_{h,z=1m}$ increases with time over the entire simulation, reaching values up to 0.48°C ($\Delta T_{h,z=1m}/T_{md}$ ~ 0.1). In contrast, for $Ro$ $O(10^{-1})$, the growth rate of $\Delta T_{h,z=1m}$ (red line in Fig. 2e) is substantially lower owing to the daily flushing of the littoral region by gravity currents. Note that $\Delta T_{h,z=1m}$ in this scenario follows the daily cycle of solar radiation, reaching its maximum of up to 0.05 °C ($\Delta T_{h,z=1m}/T_{md}$ ~ 0.01) immediately after the peak in solar radiation.

For the intermediate $Ro$ of $O(10^{-2})$ (yellow line in Fig. 2e), both of these limiting behaviors are seen. At early times, heat is retained in the shallows and the cross-shore temperature contrast $\Delta T_{h,z=1m}$ increases with time. Once a threshold contrast is reached ($\Delta T_{h,z=1m}$ ca. 0.1 °C, $\Delta T_{h,z=1m}/T_{md}$ ~ 0.025), further heating of the shallows induces an offshore transport and a quasi-

steady state becomes established. For $Ro$ $O(10^{-3})$ (blue line in Fig. 2e), the rate of change of $\Delta T_{h,z=1m}$ in time also decreases, which may also suggest the establishment of a quasi-steady state.

The simulations illustrate that horizontal temperature and density gradients increase by about an order of magnitude as $Ro$
decreases from $O(10^{-1})$ to $O(10^{-3}–10^{-2})$ and that the increasing cross-shore contrast over successive days indicates that excess heat is effectively retained in the littoral region. As a consequence, the horizontal lake circulation becomes modified by rotational effects.

## 3.2 Radial and azimuthal lake circulation

The characteristic signal of differential heating under ice, with temperature at a given depth increasing towards the littoral
region and the formation of cross-shore circulation cells, is clearly observed for $Ro$ $O(10^{-1})$ (Figs. 1b and 3a-b). Because of rotation, however, the gravity currents and the return flows are deflected towards the right in the northern hemisphere. Driven by the offshore-directed gravity currents, a cyclonic gyre develops in the lake interior (positive azimuthal velocities, $u_\theta$, in Fig. 3c); while the onshore return flow drives an anticyclonic gyre in the littoral region (negative $u_\theta$ in Fig. 3b). The gyres are governed by a three-way balance of horizontal pressure gradient, Coriolis and centripetal forces (cyclogeostrophic balance,
Sect. S1.2 in the supplementary material). The signature of the double-gyre circulation is distinguishable from the CML-averaged azimuthal velocities shown in Fig. 3d. The strength of the gyres is maximal immediately after the peak in solar radiation, when gravity currents flush the littoral region. Although weaker in strength, the double gyre circulation does not fully dissipate at night (see video S1 in the supplementary material). The weakly-dissipative nature of the gyres is in agreement with estimates of the Ekman damping time scale, $t_{Ek} = (fE_k^{1/2})^{-1}$, where $E_k$ ( $= v_z\,f^{-1}\,h_{gyre}^{-2}$) is the Ekman number, $v_z$ is the
vertical kinematic viscosity and $h_{gyre}$ is the gyre thickness. For $v_z = 10^{-6}$ m$^2$ s$^{-1}$ (see Sect. 2) and $h_{gyre}$ between 3 and 7 m (Fig. 3c), $t_{Ek}$ ranges between 3 and 7 days, which is longer than the period of the radiative forcing (less than 1 day). Flow convergence in the lake interior in the radial direction and rotation upwell isotherms at the bottom of the CML. This is illustrated by the dome shape of the isotherms at the bottom of the CML in Fig. 1b and Fig. 3a and by the positive horizontal density anomaly at the center of the lake in Fig. 1c. Isotherm upwelling explains the presence of a weaker stratification immediately below the
CML at the center of the lake (red profiles in Fig.1a and Figs. 2a-c).

Decreasing $Ro$ from $O(10^{-1})$ to $O(10^{-2})$ alters the circulation described above. The bowl shape of isotherms above 10m depth in Fig. 3e shows that temperature increases towards the littoral region. However, the radial velocity in Fig. 3f shows that the downslope cross-shore gravity currents are strongly suppressed. There, a balance is set between the Coriolis force and the pressure gradient (geostrophic balance, Eq. (S1) in Sect. S1.2 in the supplementary material). This geostrophic balance results
in the formation of an anticyclonic gyre that dominates the circulation within the CML (Figs. 3g and 3h) and restricts the cross-shore transport. For $Ro$ $O(10^{-3})$, isotherms in the littoral region get closer together (Fig. 3i) and the cross-shore transport stops (see radial velocities in Fig. 3j). As the area where cross-shore density gradients develop becomes smaller, the spatial extension of the anticyclonic gyre becomes confined to the shallow littoral region (Figs 3k and 3l). Thus, water circulation under ice is strongly affected by Earth rotation. Once a geostrophic balance is set, the circulation in the CML evolves from a combination

of radial (counter-rotating cells) and azimuthal (gyres) motions to a circulation that occurs preferentially in the azimuthal direction. This change in the lake circulation could have consequences for the heat balance of the lake interior.

### 3.3 Earth rotation modifies the interior mixed-layer heat balance

We examine the impact of rotation on the heating rate of the interior mixed layer. For this, we consider an arbitrary control volume away from the littoral region (gray cylinder in Fig. 1d) with a surface area $A_{CV}$ and extending vertically from the ice-
water interface to a depth $h_{cml}$. Neglecting interior diffusive fluxes, the heat balance in this control volume is (Ulloa et al., 2019):

$$\frac{dT}{dt} = \frac{1}{h_{cml}} (F_{ext} + F_{adv}),\tag{5}$$

where $F_{ext}$ is the sum of the solar heating rate (yellow arrow in Fig. 1d) and the diffusive heat lost towards the ice (blue arrow in Fig. 1d), and $F_{adv}$ is the sum of advective heat fluxes across the lateral and bottom boundaries of the control volume (red
and orange arrows in Fig. 1d). The daily-averaged $F_{ext}$ and $F_{adv}$ are calculated as:

$$F_{ext} = \langle \int_{z_{cml}}^{H} \frac{\partial I}{\partial t} dz - K_z \left| \frac{\partial T}{\partial z} \right|_{z=H} \rangle\tag{6}$$

and

$$F_{adv} = - \langle \frac{1}{S_{CV}} \oint T\boldsymbol{u} \cdot \hat{\boldsymbol{n}} \, dS_{CV} \rangle.\tag{7}$$

Here, the angle brackets denote both a volume and a daily averaging in the CML, $z_{cml}$ is the height of the base of the CML, $H$
is the total height of the basin (Figs. 1d and S1) and $K_z$ is the vertical thermal diffusivity. Equation (7) integrates both vertical and radial advective fluxes, $S_{cv}$ being the surface of the control volume with outward facing unit normal vector $\hat{n}$ and $\boldsymbol{u}$ being the vector velocity.

The relative contribution of advection to the total heat fluxes is, thus, $\delta_{adv} = F_{adv}/(F_{ext} + F_{adv})$. We compared this ratio, for the three $Ro$ examined, as deviations from the contribution calculated in a reference numerical experiment where the 1D approach
for deepening and warming of the CML applies, that is $\Delta\delta_{adv} = \delta_{adv} - \delta_{adv\text{-}ref}$. This reference run is identical to run 1 in Table 1, except for the lake bathymetry: the conical shape in the reference run was replaced by a cylinder of height $H$ and of surface area equal to $A_{total}$. By choosing a cylinder, we eliminated the effect of differential heating. Fig. 4a shows $\Delta\delta_{adv}$ for the three $Ro$ tested. For $Ro$ $O(10^{-2}–10^{-1})$, $\Delta\delta_{adv}$ increases over time, while for $Ro$ $O(10^{-3})$ the contribution of advection is close to zero or even takes negative values, which indicates that the net effect of advection is to export heat from the control volume towards
shallower regions. $\Delta\delta_{adv}$ reaches the highest values in the simulations with the largest $Ro$ tested. For example, from day 6 onwards, $\Delta\delta_{adv}$ with $Ro$ $O(10^{-1})$ consistently approximately triple that with $Ro$ $O(10^{-2})$. The increasing importance of advective heat transport with increasing $Ro$ is consistent with efficient flushing of the shallows and enhanced warming of the interior

CML (Figs. 2a-d) while the lack of advective flux at small $Ro$ is further indication of heat retention in the littoral region (Fig. 2e).

Advective fluxes, thus, depend on the rotation regime. This limits the applicability of the geometrical factor $G$ (Eq. (1), Ulloa et al., 2019) to predict the extra warming of the CML due to differential heating as $Ro$ decreases. To obtain $G$ (Eq. (1)), Ulloa et al., 2019) defined the length of the littoral region, $L_{shallow}$, as the distance from the lake perimeter to the location in the bathymetry where depths intersect $h_{cml}$ (Figs. 1d and S1). Adapted to a circular surface area, this corresponds to:

$$G = \left| \left( 1 - \left( \frac{R - L_{shallow}}{R} \right)^2 \right) \left( \frac{\bar{h}}{h_{cml}} - 1 \right) \right|. \tag{8}$$

Dots in Fig. 4b show that only the calculated $\Delta\delta_{adv}$ values for $Ro$ $O(10^{-1})$ are consistent with predictions using the $G$ factor (red dots in Fig. 4b), while the contribution of advection to the heat balance at the lake interior would be strongly over-predicted for $Ro$ $O(10^{-3}\text{-}10^{-2})$ (blue and yellow dots in Fig. 4b). Given that rotation controls the advection of heat to the lake interior, the characteristic length scale to define the littoral region is, however, better characterized by the Rossby radius ($Ro_R = Ro{\times}R$, Sect. S1.1 in supplementary material) whenever $Ro_R < L_{shallow}$. This leads to a new definition of $G$, here called $G_{Ro}$, expressed

as a function of the Rossby radius and, specifically for our bathymetry, as (see details in Fig. S1 and Sect. S1.3 in the supplementary material):

$$G_{Ro} = \left| \left( 1 - \frac{(R - L_{shallow})^2}{(R - L_{shallow} + Ro_R)^2} \right) \left( \frac{\overline{h_{Ro}}}{h_{cml}} - 1 \right) \right| \tag{9}$$

where $\overline{h_{Ro}}$ is the average depth of the littoral region within a distance $Ro_R$ from the lake interior (Fig. S1). Note that for $Ro_R \geq L_{shallow}$, $G_{Ro} = G$. By using $G_{Ro}$ when $Ro_R < L_{shallow}$, the trends in the contribution of advection of heat from the littoral region

to the heat balance at the lake interior are better represented, especially for the scenario with the lowest $Ro$ (blue squares in Fig. 4b).

## 4. Discussion

### 4.1 Advection of heat from the littoral regions decreases with decreasing $Ro$ (big lakes and/or high latitude lakes)

In this study, we show that, subjected to the same radiative forcing, the under-ice warming of lakes is strongly modulated by

the combination of lake bathymetry and latitude (i.e., intensity of Earth rotation). Once solar radiation is able to penetrate the ice and heat the water below it, under-ice convection becomes the main driver of lake circulation and mixing (Kirillin et al., 2012). Convective cells impinge on the stratified layer below, entraining water from below and deepening the CML. This process is often viewed as a one-dimensional process (Farmer, 1975). However, the presence of a bathymetry with sloping boundaries implies that as the CML becomes deeper in the lake interior, convective deepening is constrained by the bottom in

the shallower regions. With a smaller volume to heat up, these shallower areas warm faster than the lake interior and a horizontal density gradient develops. If this excess heat is advected towards the lake interior, the warming and deepening of

the CML in the lake interior speeds up (Ulloa et al., 2019; and Fig. 2a-d). The advection of heat from the littoral region to the lake interior is, however, constrained by Earth rotation. Our findings show that the rotation regime plays a fundamental role in the heating rate of the CML and in the cross-shore and vertical heat distribution (Figs 2-3). As the rotation increases, the vertical and radial heat exchanges are dramatically inhibited (Figs. 3-4a). In fact, the warming and deepening rates of the interior CML for the scenario with the smallest Rossby number ($O(10^{-3})$) examined, match the results of the reference simulation, which lacks differential heating and lateral neat heat transport. For this scenario, the density difference between the littoral region and the lake interior increases over time until the end of the simulation (Figs. 2e). This indicates that heat is effectively retained in the littoral region (see also Fig. 3k). Consistent with the results of our simulations, the new $G_{Ro}$ (Eq. (9)) geometrical parameter that adopts $Ro_R$ as the representative length scale when $Ro_R < L_{shallow}$, predicts a decrease in the lateral advection of heat as $Ro$ decreases.

## 4.2 Lake circulation

Differential heating and Earth rotation set up a circulation characterized by the formation of gyres within the CML (Fig. 3 and Fig. 5). The double-gyre circulation (Fig. 5a) for $Ro$ $O(10^{-1}$–$10^0)$ is consistent with past field-based work on ice-covered lakes. Forrest et al. (2013) conducted CTD measurements mounted on an autonomous underwater vehicle and observed a density distribution in the interior of Pavilion Lake ($50°$ N, $R \sim 0.4$ km and $A_{total} = 5$ km$^2$) that suggested the existence of a cyclonic gyre for $Ro$ $O(10^{-1})$ (recalculated from Forrest et al. (2013) using $L = R$). Kirillin et al. (2015) conducted CTD profiles across lake Kilpisjärvi ($69°$ N, $R \sim 1.5$ km and $A_{total} = 37$ km$^2$) and ADCP measurements at the littoral region when $Ro$ was $O(10^{-1})$. With the former data, they observed warm "upwelling" at the lake center, which could be indicative of a strong cyclonic gyre in the lake interior; and with the latter, they measured radial velocities of ~3-5 cm s$^{-1}$ and the presence of an anticyclonic gyre in the littoral region with azimuthal velocities of 2-4 cm s$^{-1}$. A double-gyre circulation was also reproduced in numerical simulations (Huttula et al., 2010) of early winter conditions in Lake Pääjaarvi ($61°$ N), when under-ice circulation was dominated by the input of heat from the sediment (Winter I, Kirillin et al., 2012) instead of by radiatively-driven convection (Winter II, Kirillin et al., 2012). Also when the input of heat from the sediment dominated lake circulation, Likens and Ragotzkie (1966) injected radioactive tracers near the center and in the littoral region in Tub Lake ($45°$ N, $R \sim 50$ m and $A_{total} = 8.4 \times 10^{-3}$ km$^2$) and detected the presence of a double-gyre circulation when $Ro \sim 0.1$ (calculated with their reported horizontal velocities of $3.5 \times 10^{-4}$ – $4 \times 10^{-4}$ m s$^{-1}$). The central cyclonic circulation had already been detected in this same lake by Likens and Hasler (1962) in a previous winter, suggesting that this azimuthal circulation pattern is recurrent during the ice-on season in the lake. Note, however, that for the ageostrophic regime to display a double-gyre circulation, the cross-shore cell circulation should exist over a significant fraction of an inertial period ($\sim f^{-1}$). In our simulations, $I_0$ is close to its daily maximum for ca. 6 h, but cloudiness or mountain shading could decrease this duration in real settings. When $Ro \gtrsim 10^{-1}$, the horizontal heat transport is then accomplished by the ageostrophic components of the flow (downslope gravity currents). This cross-shore circulation might be considered analogue to the atmospheric Hadley cells, as reproduced in laboratory rotating-tank experiments (e.g., Fultz et al., 1959).

As *Ro* decreases, the circulation in the lake becomes geostrophic (Fig. S3), the daily cross-shore cells are rotationally suppressed and, because of the horizontal density gradient, an anticyclonic gyre (in the northern hemisphere) with velocities decreasing with depth must develop (Eq. (S1) and Figs 5b). With decreasing *Ro*, the spatial extent of this anticyclonic gyre is progressively confined to the littoral region (Fig. 5c). To our knowledge, this circulation has not been reported in the field at times of under-ice radiatively-driven convection. The lake-wide anticyclonic circulation in Fig. 5b would be consistent with

the inferred lake-wide anticyclonic gyre reported by Rizk et al. (2014) at a time when circulation in lake Pääjärvi was dominated by a lateral gradient in the heat flux from the sediment (Winter I) and *Ro* was $O(10^{-3}\text{-}10^{-2})$. Nonetheless, Welch and Bergmann (1985) reported radial velocities of $1\times10^{-4}$ m s$^{-1}$ during Winter I in Methane Lake (63°N, $R \sim 100$ m and $A_{total}$ $< 0.1$ km$^2$) that would lead to estimates for *Ro* of $O(10^{-3}\text{-}10^{-2})$. By adding a dye (rhodamine) in a point in the littoral region and close to the lake bed, they detected the presence of density currents flowing offshore and no sign of gyre formation. This

would be contrary to the expected radiatively-driven lake circulation in the geostrophic regime as presented in this study, and suggests that (1) other processes could be at play during Winter I or that (2) the radial velocity magnitude, and thus *Ro*, was underestimated by the authors. The latter is possible given that Welch and Bergmann (1985) used dye concentrations to estimate $O(10^{-4})$ m s$^{-1}$ radial velocities in the lake. Due to a different distribution of the pressure field (decreasing towards the littoral region in the under-ice differential heating case), the sense of the azimuthal circulation in Figs. 3g,h and Fig. 3k,l is,

for example, opposite to that observed inshore of thermal bars (ice-free period), where a cyclonic gyre develops (e.g., Huang, 1972; Malm et al., 1993). The basin-scale gyre circulation in Figs. 3g,h and Fig. 3k,l and its sense of rotation are, however, consistent with the Rossby wave regime reported in laboratory studies (rotating cylinder and annulus) mimicking the mid-to-high-latitude atmospheric circulation (e.g., Condie and Rhines, 1994; Fultz et al., 1959; Sommeria et al., 1989). Water temperature in the rotating tanks is above $T_{md}$ and the sense of the gyre rotation is anticyclonic when the heating and cooling

sources are provided at the center and the tank rim, respectively. The sense of rotation is reversed (cyclonic gyre) when the heating and cooling sources are exchanged. Within the Rossby regime, vortices and waves develop (Fig. 3h) and as *Ro* decreases, the wave lengths decrease and the gyre circulation is concentrated into jets that meander in the radial direction and could finally break (Condie and Rhines, 1994; Read et al., 2015; Smith et al., 2014). The presence of waves and/or vortices as in the scenario with *Ro* $O(10^{-2})$ (Fig. 3h) is typical of transitional regimes (Fultz et al., 1959) and when they develop, the center

of the anticyclonic gyre is not static in time but fluctuates laterally (video S1). This lateral displacement favors the existence of pulses of gravity currents from the littoral region to the lake interior, leading to a higher contribution of advection in the heat balance of the lake interior. The lateral fluctuations of the gyre cannot be accounted for in the geometrical factor $G_{RO}$ and explains the under prediction of the contribution of advection for *Ro* $O(10^{-2})$ in Fig. 4b (note that the yellow squares in Fig. 4b are above the 1:1 line).

**4.3 Implications**

This study reveals that for a given radiative forcing and latitude, as a result of differential heating, a temperature mooring deployed at the deepest point of an ice-covered lake will record a higher warming rate if shallow regions contribute

substantially to the lake volume. Thus, for a given lake surface area, a lake with more vertical walls would tend to warm at a slower rate than a bowl-shaped lake with gentle slopes. However, if we could virtually move these two same lakes to higher

latitudes, that mooring could potentially record the same warming rate in the two lakes. That is because the contribution of the littoral region to the warming of the lake interior under ice is strongly dependent on its size and its latitudinal location, both of which condition the importance of Earth rotation (parameterized by $Ro$) in driving lake water circulation. The range of $Ro$ values investigated here lies within the natural range of variability of $Ro$ that characterize lake circulation driven by under-ice differential heating in lakes on Earth (Figs. 5d-e). This means we are covering ice-covered lakes from the Himalayas and the

Tibet Plateau (~30° N) to high-latitude polar regions. The bias in the distribution of lake size towards smaller lakes (more than 85% of lakes on Earth have surface areas smaller than 1 km$^2$, Messager et al., 2016) and the bias in the potential for ice cover formation (e.g. Sharma et al., 2019) and the distribution of lakes on Earth towards mid to high latitudes (> 45° N, Fig. 5b) indicates that the ageostrophic regime with $Ro$ O($10^{-1}$) should be common under ice as shown by the mean (red lines) values in Fig. 5d. The geostrophic regime, $Ro < $ O($10^{-1}$), lies outside the interquartile range (gray bars in Fig. 5d), but is also naturally

occurring in lakes in almost all latitudes (Fig. 5d). The retention of heat in the littoral region in the geostrophic regime could potentially reinforce ice melting near the shoreline. This is suggested in our simulations where with $Ro$ O($10^{-3}$) the diffusive flux towards the ice is on average 23 % larger in the littoral region than at the center of the lake (Profiles P1 and P2 in Fig. 1c). However, heat retention in the littoral region does not necessarily imply an increase in the diffusive flux towards the ice. The diffusive flux, $K_z \, \partial T/\partial z$, depends not only on the temperature in the shallow region, but also on the thickness of the

diffusive boundary layer, which determines the value of $\partial T/\partial z$. Due to the retention of heat in the littoral region, water temperature there could potentially reach values $\geq T_{md}$. This would lead to the development of a stable stratification in the littoral region and to the suppression of convection that, in contrast, would continue in the lake interior. This could have implications for early formation of thermal bars and/or contribute to the formation of moats (e.g., Nolan et al., 2003).

The numerical experiments described herein intend to provide a general characterization of the respective contribution of

lake bathymetry and rotation on the warming of lakes under the ice; however, site-specific conditions will determine the actual response of a given lake. For example, although basin or sub-basin scale gyre formation has been reported to occur in lakes with bathymetries departing from the bowl-shaped one used in this study (e.g., Forrest et al., 2013; Kirillin et al., 2015), bathymetric effects could prevent gyre formation, or, by contrast, lead to the development of more complicated patterns (for example, multiple horizontal gyres, as in Huttula et al. (2010)). Also, the initial conditions could vary among

lakes (e.g., Yang et al., 2020b). Although there are examples in the literature of lakes reaching values close to $T_{md}$ near the lake bottom (e.g., Bengtsson and Svensson, 1996; Cortés and MacIntyre, 2020; Forrest et al., 2013; Malm et al., 1998), other lakes reach temperatures of only 2-3 degrees at the bottom of the lake (e.g., Bouffard et al., 2016; Yang et al., 2020a). The radiative forcing conditions could also depart from those in this study. The radiative flux could vary spatially (e.g., Malm et al., 1997) and the daily radiative cycle could vary over time (e.g., Bouffard et al., 2016). The initial and forcing conditions

(magnitude and time evolution) will influence the deepening rate of the CML, and/or the strength of differential heating and density currents, and will also determine the magnitude of $Ro$.

Depending on the magnitude of the forcing, and the thickness of the convective mixing layer $h_{cml}$, the Rayleigh number, which quantifies the strength of buoyancy over diffusion, $Ra = g\alpha I_0\, h_{cml}^4/(v_z\, K_z^2)$, can easily reach magnitudes of $\sim O(10^{15})$ or even higher if we consider the horizontal convective length as the relevant scale. Here $\alpha$ is the thermal expansion coefficient. These magnitudes are difficult to achieve in laboratory and numerical experiments (King et al., 2009). There have been significant advances in in-situ measurements over the last years that facilitate more robust measurement of turbulence and mixing properties (Bouffard et al., 2019). Thus, ice-covered lakes might provide unique observations to test theoretical advances on the heat and mass transfer in rapidly rotating convection in fluid systems of planetary scale (e.g., Julien et al., 2012; King et al., 2009). Large and high-latitude ice-covered lakes provide, therefore, unique opportunities to investigate strong convective regimes at low ($\ll 1$) Rossby and Ekman numbers and with Prandtl numbers $\approx 10$.

## 5. Conclusions

Penetrative radiation affects the under-ice melting rate by regulating the water-to-ice heat transfer and is often modeled as a one-dimensional process. Our results show, however, that lake bathymetry and latitudinal location also affect the rate of warming of ice-covered lakes. Lake bathymetry induces a relative excess heating of shallow waters. The transport of this excess heat to the lake interior depends on the intensity of Earth rotation and determines lake warming rates and the horizontal distribution of heat. This study stresses that accounting for the shape and size of the lake basin and its latitudinal location is essential for global estimations of lake ice cover that take into account the warming rates and the distribution of heat in the water column.

### Code availability

The MITgcm code is publicly available at http://mitgcm.org

### Data availability

The data displayed in the figures, temperature profiles at P1 and P2, time series of cross-sectional data and the input files to run the simulations are available for download at Zenodo (http://doi.org/10.5281/zenodo.4027393). The complete time series of the 3D model outputs used in this study can be requested from the corresponding author.

### Author contribution

C.L.R has run the simulations, conducted the bulk analysis and led the writing of the manuscript. The other co-authors have equally contributed to the study design, interpretation of the data and editing of the manuscript.

**Competing interests**

The authors declare that they have no conflict of interest.

**Acknowledgments**

This work was supported by the Swiss National Science Foundation (project Buoyancy driven nearshore transport in lakes, HYPOlimnetic THErmal SIphonS, HYPOTHESIS, reference 175919). K.B.W. acknowledges support from the U.S. National Science Foundation, Physical Oceanography, Award OCE-1657791.

We thank the two anonymous reviewers for their thorough comments and suggestions.

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

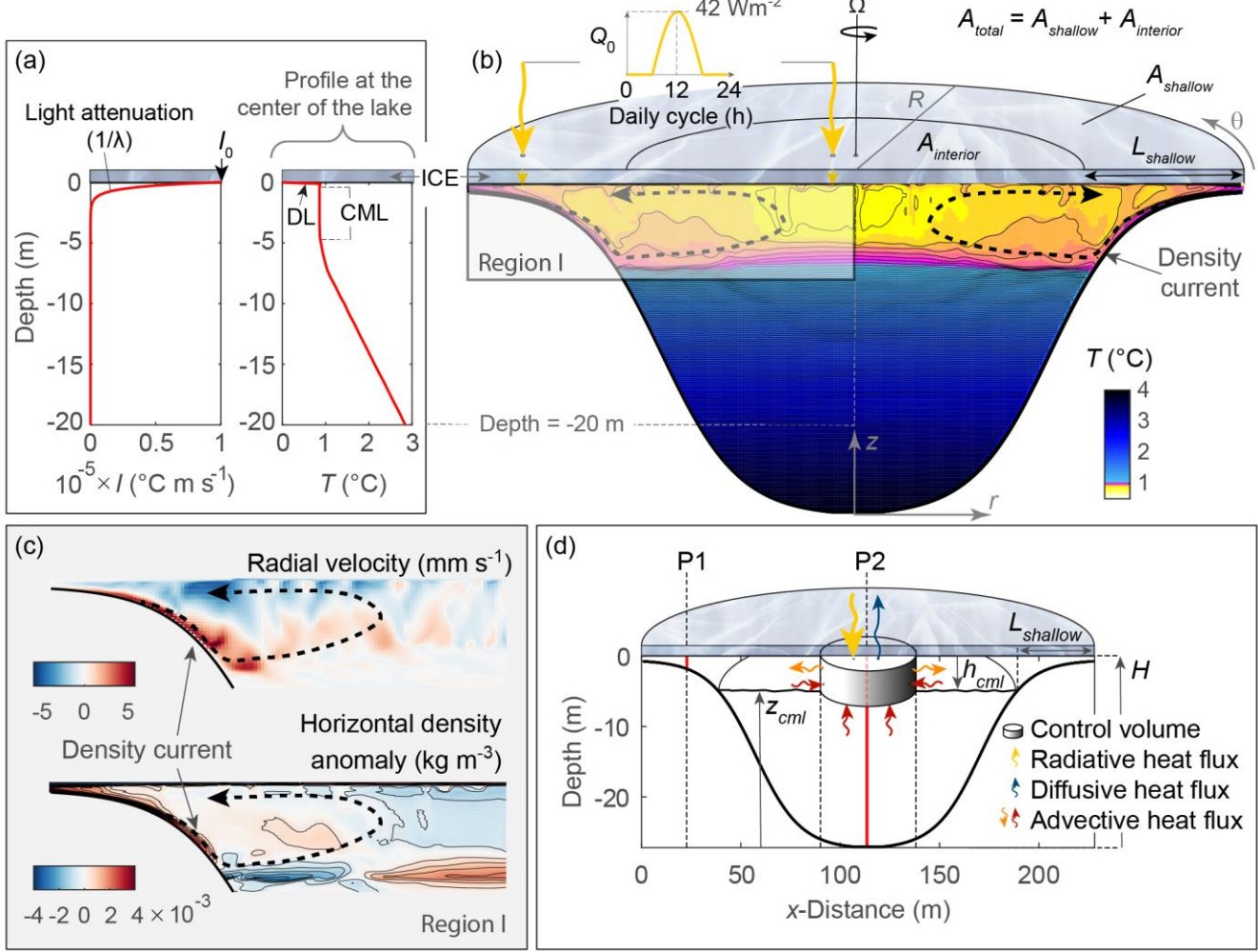

**Figure 1: Schematic of the modeled lake forcing and under-ice differential heating. (a) Variation in depth of the radiative forcing (I) and the resulting temperature profile at the lake center, characterized by the formation of convective mixed layer (CML) immediately below the thin diffusive layer (DL). (b) Daily cycle of the radiative heat flux and snapshot of the temperature field in a cross-section showing gravity currents developing in the littoral region as a result of differential heating for $Ro$ O $(10^{-1})$. The isotherm spacing is 0.02°C. (c) The signature of these gravity currents can be also detected by positive radial velocities (positive towards the lake interior) and positive horizontal density anomalies (deviations from the basin-average density profile). (d) Schematic of the mixed-layer heat balance at the lake interior and location of profiles P1 and P2 (red lines), ~91 m apart, used in Fig. 2.**


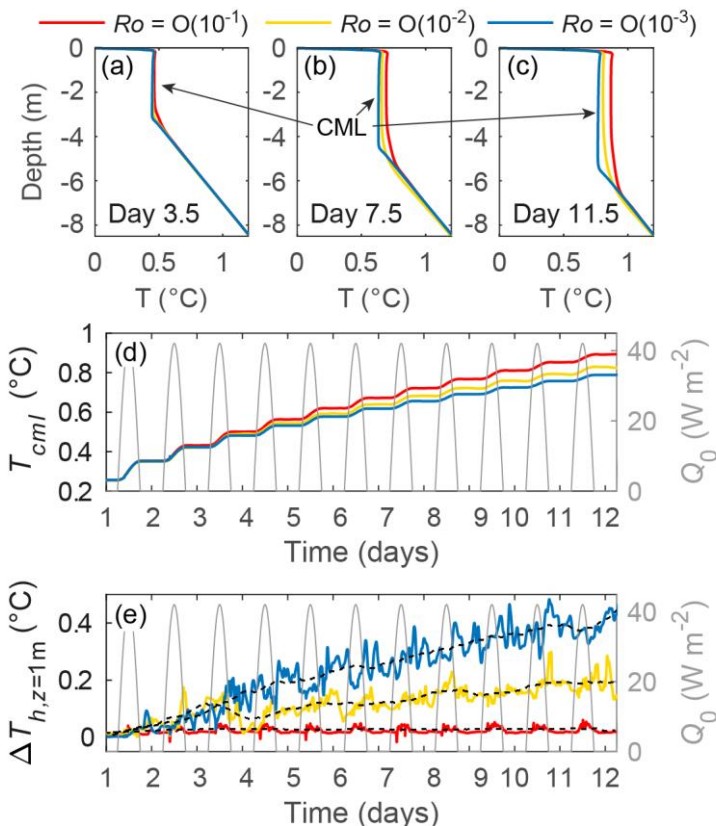


**Figure 2: Warming of the CML in the lake interior and differential heating. (a-c) Laterally averaged midday temperature profiles on given days in the lake interior (lake volume below the surface area of the control volume in Fig. 1d). (d-e) Time evolution of (d) temperature in the interior of the CML (average in the control volume in Fig. 1d) and (e) the horizontal temperature gradient at 1-m depth between the littoral region and the lake interior (Profiles P1 and P2 in Fig. 1d). Black dotted lines in e show the 24-h moving**
**averages.**

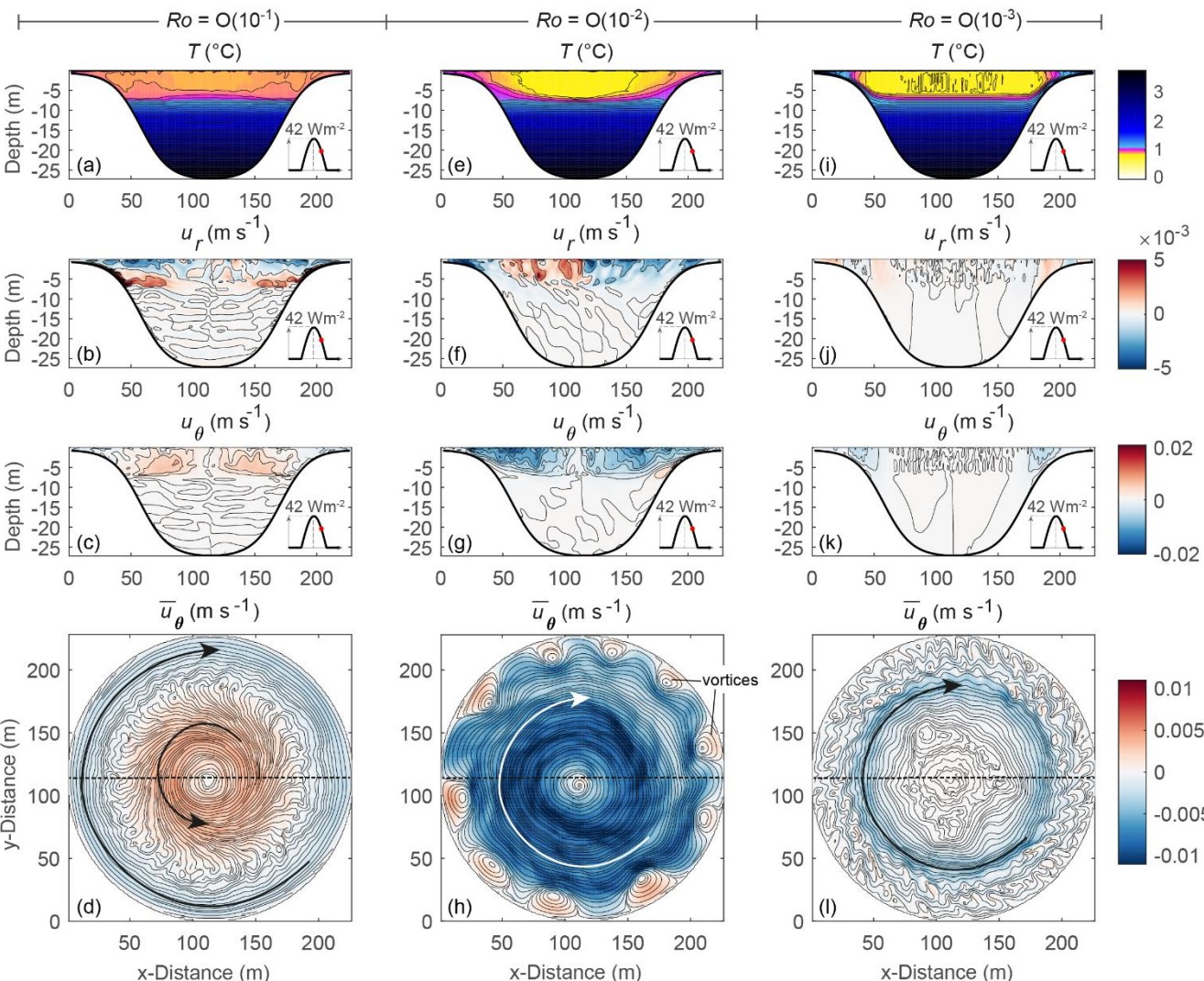

**Figure 3: Snapshots of the thermal and velocity structure for different *Ro*. Snapshots from day 11.6, just after peak insolation, showing the simulated cross-sectional (a, e, i) temperature (0.02°C isotherm spacing), and (b, f, j) radial and (c, g, k) azimuthal velocities (0.002 m s⁻¹ isovel spacing); and depth-averaged (d, h, l) azimuthal velocities and flow streamlines at depths ≤ $h_{cml}$ . Radial and azimuthal velocities are positive towards the lake interior and for cyclonic circulation, respectively. Black dashed lines in (d, h, l) show the location of the cross-section displayed in (a-c, e-g and i-k). Results for runs with (a-d) *Ro* O(10⁻¹), (e-h) *Ro* O(10⁻²) and (i-l) *Ro* O(10⁻³).**

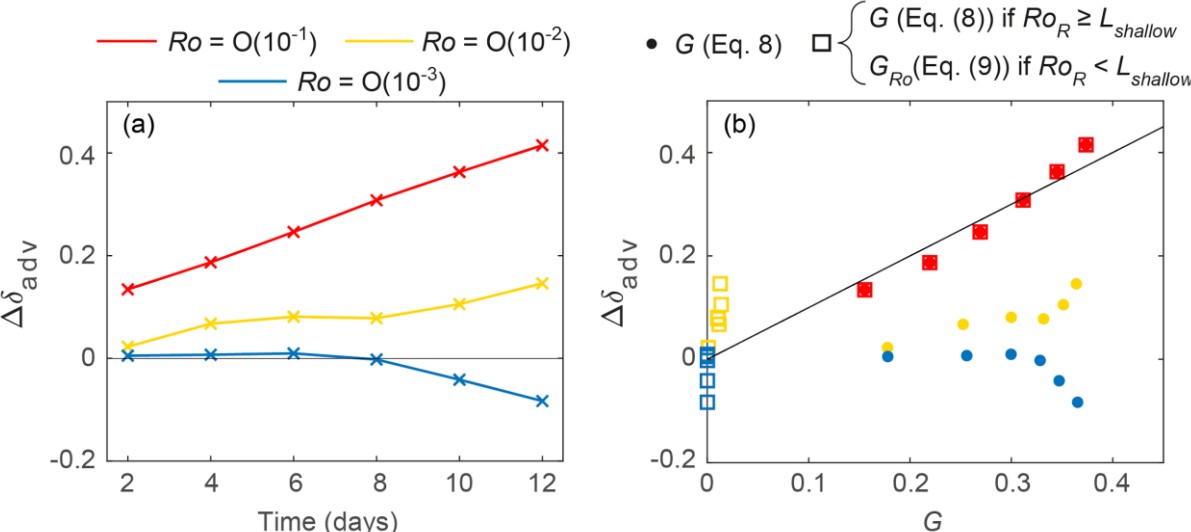

**Figure 4: Contribution of advection to the total heat fluxes into the lake interior. (a) Time evolution of the simulated $\Delta\delta_{adv}$. (b) Simulated versus predicted contribution of advection based on the geometrical factors $G$ (Eq. (8)) and $G_{Ro}$ (Eq. (9)). Each point in (a-b) represents the average value over the previous two days. The black line in (b) shows the 1:1 relationship.**

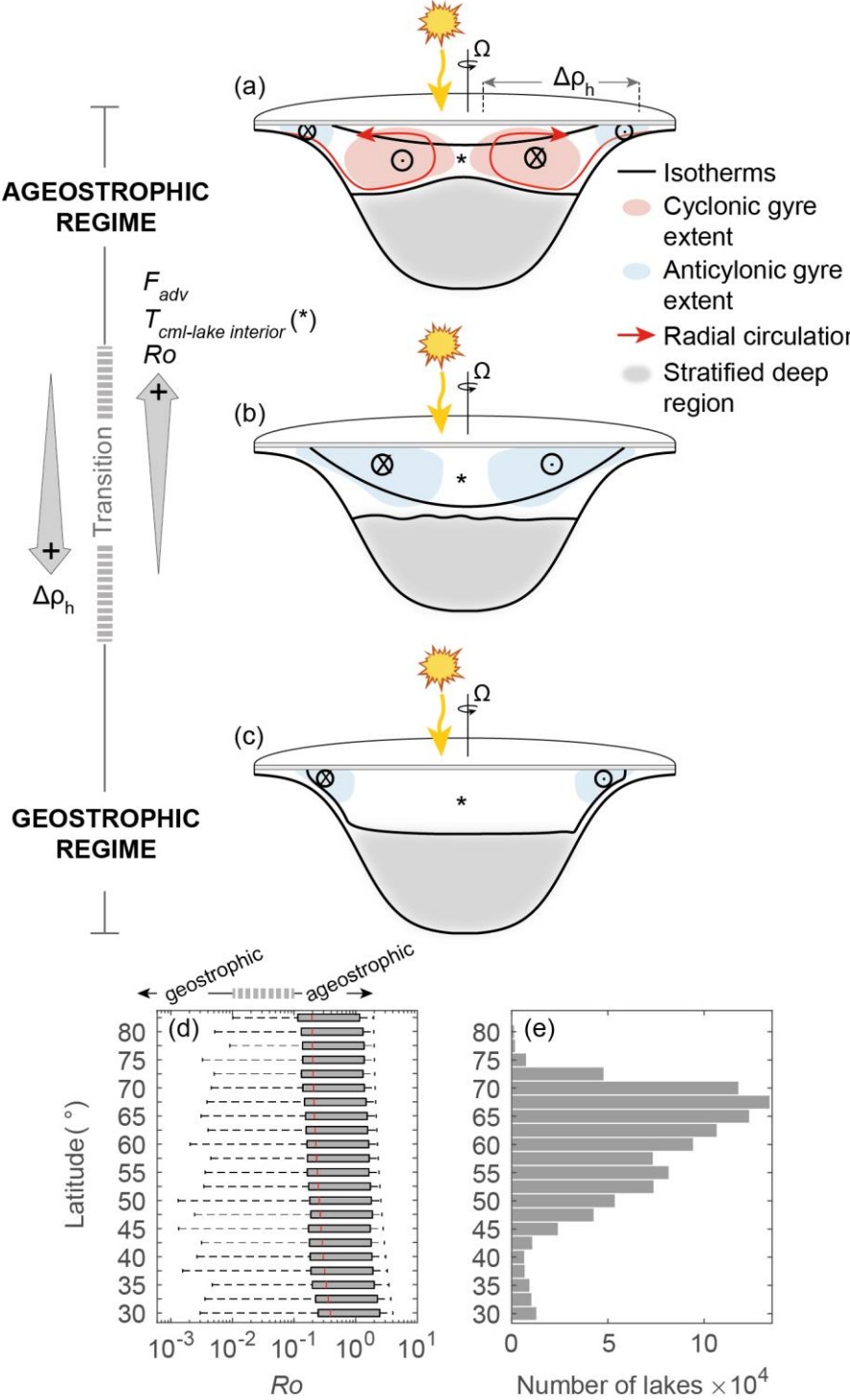

**Figure 5: Conceptual model for under-ice circulation induced by differential heating. (a-c) Conceptual model for the ageostrophic and geostrophic regimes under ice. (d) Boxplot of *Ro* values and (e) distribution of lakes used for the calculations of *Ro* for latitude intervals of 2.5°. The HydroLAKES database (Messager et al., 2016) was used to construct panels d-e (see details in Sect. S1.4 in the supplementary material). Latitude in (d-e) is expressed in absolute values. Red lines and gray rectangles in (d) show the median value and the interquartile range, respectively**

**Table 1: Test cases. Value of the Rossby numbers (*Ro*) in our simulations**

| Run | *Ro*\* | Bathymetry |
|---|---|---|
| Run 1 | 0.230 ± 0.168 | Conical (Fig. 1B) |
| Run 2 | 0.019 ± 0.011 | Conical (Fig. 1B) |
| Run 3 | 0.001 ± 0.001 | Conical (Fig. 1B) |
| Reference | - | Cylinder |

\*Average value ± standard deviation after 12 radiative cycles (Fig. S2c).