# Peer review of "Bathymetry and latitude modify lake warming under ice"

_Hydrology and Earth System Sciences, 2020_

## Referee Comment (RC1) · Anonymous Referee #1 · 18 Nov 2020

In this paper the authors consider how late winter solar radiation driven convection may be controlled by the relative strength of Coriolis force. As many as half the lakes in the world freeze and are located at high latitudes where Coriolis forces could be important. However it is very hard to do field work under these lakes, and while the process of solar driven convection is somewhat well known, how Coriolis forces interact with lateral temperature gradients to drive basin wide circulation is not well known. This paper represents an important first step to address some of these issues. The paper is well written, and my comments below are mainly about putting this new work into context of existing literature of Coriolis effects on convection. Addressing these comments does not require any changes to figures, rather just some careful thought to some of text so should be somewhat straight forward and will make the paper stronger.

[Figure]

1) I think it would be very helpful for authors to emphasize some caveats and qualifications to the generality of their study. What might happen in the many lakes that are long and narrow? Do you expect multiple gyres? Could the authors comment on whether the Rossby number is time dependant in a real lakes, due to increases in buoyancy forcing as the length of the day increases? This might mean change in sense of circulation patterns over the end of winter . How typical is the strong stratification that is as warm as 4C at bottom? Many large but relatively shallow lakes are less stratified and maybe be 2C or 3C at base (see https://aslopubs.onlinelibrary.wiley.com/doi/abs/10.1002/lno.11543 as one example). In these less stratified lakes you are much more likely to see gravity currents going to base of lake, so are circulation patterns are possibly different?

2) I was also wondering if the title might be slightly qualified? Rather than "Latitude and geometry", I'd suggest order be "geometry and latitude", as I think geometry is more important. You find the vast majority of dimictic lakes from about 40 to 70oN (Northern America and Northern Europe) where f varies from 0.935 to 1.367 10-4sˆ-1. So most of the variation in Rossby number between lakes is not primarily due to latitude, but rather their scale (and possibly magnitude of radiation which is indirectly also related to latitude).

3) I think you need to qualify that statement on line 20 that "Yet, as we move from planetary to smaller-scale systems, the importance of rotation in affecting convective processes remains overlooked. This is the case in lakes." I think you want to change or qualify the word "overlooked". In terms of convection there have many studies of the "spring thermal bar" where heating of lakes that are below 4oC drives a radial geostrophic flow near the shore, ( https://en.wikipedia.org/wiki/Thermal_bar ). This seems to be very closely related physics, so It would be worth reminding the reader of the connection between your study and this well known process in larger lakes (which is probably at low Ro end of your simulations). This is a well known example of Coriolis limiting heat transfer from edges to interior.
What I am confused about is that all observations and models thermal bar suggest you'd get an anti-clockwise flow (cyclonic) near the shore in Northern Hemisphere. I think this is opposite however to what is shown in low Rossby number case for Figure 3. There are many theoretical papers from 1970s on thermal bar, one example is https://doi.org/10.1080/03091927208236071 Huang, Joseph Chi Kan. "The thermal bar."Âă Geophysical Fluid DynamicsÂă 3, no. 1 (1972): 1-25.

This article mentions that "The results show a dominant meridional cyclonic flow along the perimetric edge of the lake and an anticyclonic flow in the middle portion of the lake.Âă" I am confused what is the key difference between that classic field observation and your simulations for low Rossby number? I may be confused here, but if the sense of circulation is fundamentally different it would be useful to explain what is the key difference in setting up your simulations.

4) There is a recent paper by Jazi Davarpanah et al. (2020) on rotating gravity currents using the Coriolis facility in Grenoble that goes into great detail on Rossby number effects and is a better reference than Wells, 2009 on line 32

https://agupubs.onlinelibrary.wiley.com/doi/abs/10.1029/2019JC015284

This study would also qualify your statement on line 149-150 that "When R/U > f gravity currents are affected by Earths rotation" - rather the large Grenoble experiments found that there is a gradual transition in gravity current dynamics that starts at Rossby number greater than 1,

5) A number of studies in last 40 years have studied convection in rapidly rotating "dishes" - as an analog to understanding zonal jets in gas giants like Jupiter. Physically one might expect that these should have same or similar circulation patterns as your low Rossby number simulations (although they lack stratification). Hence it would be worth briefly commenting to what degree the circulation patterns look similar or different to your Figure 3. I appreciate the experiments below are not stratified, but for many physicists these would be the rotating experiments they are familiar with.

One recent example from Grenoble os

Read, P.L., Jacoby, T.N.L., Rogberg, P.H.T., Wordsworth, R.D., Yamazaki, Y.H., Miki-Yamazaki, K., Young, R.M., Sommeria, J., Didelle, H. and Viboud, S., 2015. An experimental study of multiple zonal jet formation in rotating, thermally driven convective flows on a topographic beta-plane. Ăă Physics of Fluids, Ăă 27(8), p.085111.

Before computer simulations were easier this type of system was also used in some high profile papers in 1980-1990s, see figure 2 in

Condie, Scott A., and Peter B. Rhines. "A convective model for the zonal jets in the atmospheres of Jupiter and Saturn." Ăă Nature Ăă 367, no. 6465 (1994): 711-713.

Sommeria, J., Meyers, S.D. and Swinney, H.L., 1989. Laboratory model of a planetary eastward jet. Ăă Nature, Ăă 337(6202), pp.58-61.

also used to think about the modes of convection driven circulation under sloping geometry of Lake Vostok - see for instance the change from differential heating to columnar vortices in https://agupubs.onlinelibrary.wiley.com/doi/full/10.1029/2007GL032162

6) There are a few more key studies on ice covered lakes that can be compared directly to the simulations. In particular in old studies on Tub lake, the scale and geometry looks like about exactly scale as in the present student study. The lake is symmetric and has a profile from 0-4C, so is probably as similar as you could find, so a good question is whether the sense of circulation in studies by Likens is the same? They inferred basal heating was very important (as have other under ice studies during winter I.) but I feel this should be somewhat similar to radial differences in temperature gradients in winter II.

LIKENS, G. E., AND A. D. HASLER. 1962. Movements of radiosodium (NaZ4) within an ice-covered lake. Limnol. Oceansgr. 7: 48-56.

LIKENS, G. E., AND R. A. WACOTZKIE. 1965. Vertical water motions in a small ice-covered lake. B. Geophys. Res. 70: 2333-2344.

Likens, G.E. and Ragotzkie, R.A., 1966. Rotary circulation of water in an ice-covered lake: With 6 figures and 1 table in the text. Internationale Vereinigung für theoretische und angewandte Limnologie: Verhandlungen, 16(1), pp.126-133.

Another old paper shows possible sinking near boundaries, consistent with your observations

Welch, HE, & Bergmann, MA (1985). Water circulation in small arctic lakes in winter. Canadian Journal of Fisheries and Aquatic Sciences , 42 (3), 506-520.

I hope all these comments are helpful in providing some more context to your interesting simulations.

―――――――――――――――――――

---

## Referee Comment (RC2) · Anonymous Referee #2 · 9 Dec 2020

By application of a circulation model in an idealized domain mimicking the heating of an ice-covered lake of irregular morphometry by solar radiation, the authors arrive at an insightful demonstration of the rotation effects on the radial density flows produced by differential heating between shallow and deep lake areas. Rotational gravity flows are widespread in geophysical fluids and an advance in their quantification makes a valuable contribution to earth and planetary fluid dynamics. The ice-covered lakes represent rare natural examples, where these flows can be observed and investigated in detail at their whole range of scales, undisturbed by more energetic flows, usually persisting in open water, oceans, or the atmosphere. In that sense, the authors discuss an intriguing problem, of interest for a wide research community. The modeling methods are relevant, and the results are presented in a well-structured way.

I had the opportunity to read the previous comment and generally share the concerns of the Reviewer1: my major criticism refers to the weak connection of the model configuration to the real conditions met in lakes and, as a result, misleading, superfluous, and over-generalized conclusions made by the authors.

Instead of nondimensionalization of the problem with regard to the rotational forces prior applying a numerical model, the authors voluntary choose the domain dimensions of $\mathcal{O}(10^2)$ m and vary the Coriolis parameter within 2 (!) orders of magnitude. It is left for the reader's inspiration to imagine where on Earth $f = \mathcal{O}(10^{-2})\ s^{-1}$ can be observed (Line 139, Table 1). By using *a postreriori* re-scaling based on the Rossby number (Eq. 4, Line 148), a conclusion can be drawn that the ageostrophic regime ($Ro = \mathcal{O}(10^{-1})$, Fig. 3, first column), similar to that described by Ulloa et al. (2019), can be found only in small ponds with an area of several ha. In lakes with characteristic length scales of $\mathcal{O}(1)$ km ($Ro = \mathcal{O}(10^{-2})$, Fig. 3, second column) and longer ($Ro = \mathcal{O}(10^{-3})$, Fig. 3, second column), the shallow near-shore areas are effectively decoupled from the lake interior by rotation. The modeling results do not however provide a final proof for the importance of differential heating even in small ponds: they are typically much shallower than the modeled domain and have the background mixing intensities higher than those adopted in the model (Lines 108-111).

Herewith, the following outcomes of the study must be made clear: 1. For *the vast majority of ice-covered lakes*, differential heating does not contribute to the vertical mixing in the lake interior. 2. The previous findings of Ulloa et al. (2019) must be reconsidered taking into account the new results. 3. All variations of the Rossby number should be clearly related to corresponding variations in lake horizontal dimensions. Any mentioning of latitudinal effects should be removed, since for all seasonally ice-covered lakes $f = \mathcal{O}(10^{-4})\ \mathrm{s}^{-1}$.

Other remarks:

L68 The geometrical factor $G$ (Eq. 1 and Eq. 8) is of little predictive power as long

as the hypsometry (the shape of the basin) is not included. When derived in a strict way, $G$ incorporates a "shape factor" $S = 0..1$, which is found as an integral $S = \int_0^1 D(x)dx$, where $D(x) = 0..1$ is dimensionless depth, $x = 0..1$ is the relative distance from the shore to the lake center. For vertical walls $S = 0$, for linear slope $S = 0.5$, for the typical "bowl"-shaped lake $S \approx 0.3$, and for the authors' $tanh$-approximation $S \approx 0.6$. Hence, application of uncorrected $G$ to different basin shapes can lead to $\geq 2$ times differences in the result. Removal of $G$ and related discussion is *strongly* recommended unless the basin shape is incorporated in the scaling.

L131 $1/\lambda = 2.5$ m$^{-1}$ ($< 1$ m Secchi depth) is rather turbid than moderately clear and is not typical for the majority of ice-covered lakes. Would the differential heating increase in more transparent waters? How the transparency affects the rotation effects? Make it clear in the text.

L202-206 The geostrophic balance does not hold true in the bottom boundary layer (BBL), and the Coriolis effect on bottom-slope currents is strongly reduced by bottom friction. How good is BBL is reproduced by the model?

L314 Avoid the term "fjord-type" lakes, because the effect of the non-unity horizontal aspect ratio was not investigated in the study,

L320 "...Peruvian Andes ...": Any example of a seasonally ice-covered lake at latitudes below $15^o$? A lake at $20^o$ lat or lower can develop ice cover only at altitudes where liquid water is extremely rare. The whole discussion on the latitudinal variability is vague and should be rethought in terms of lake size (see above).

L321-323 The sentence is unsupported and — to be straight — wrong and misleading. Has to be removed.

L330-332 It is quite an interesting point deserving more discussion in view of the presented results. If the littoral water temperatures reach the maximum density point (TMD),

but the lake interior stays colder than TMD, the ice cover will quickly melt over the shallows, forming the well-known "moats". As long as the rest of the lake stays ice covered, water temperatures in these open areas will be close to TMD without long-lasting stable stratification. "Moat" formation has been traditionally referred to terrestrial heat fluxes; the role of heat capture by rotation was never considered in this context but deserves a closer discussion.

Presentation in form of a HESS publication of the otherwise well-performed and insightful study to a wider community can be recommended *only* after resolving the above-mentioned issues.

---

## Author Comment (AC1) · 7 Jan 2021

**Response to Referee #1**

*In this paper the authors consider how late winter solar radiation driven convection may be controlled by the relative strength of Coriolis force. As many as half the lakes in the world freeze and are located at high latitudes where Coriolis forces could be important. However it is very hard to do field work under these lakes, and while the process of solar driven convection is somewhat well known, how Coriolis forces interact with lateral temperature gradients to drive basin wide circulation is not well known. This paper represents an important first step to address some of these issues. The paper is well written, and my comments below are mainly about putting this new work into context of existing literature of Coriolis effects on convection. Addressing these comments does not require any changes to figures, rather just some careful thought to some of text so should be somewhat straight forward and will make the paper stronger.*

We thank Referee #1 for his/her encouragement and very interesting suggestions on how to improve the manuscript. We present below how we will implement the changes suggested by Referee #1.

*1) I think it would be very helpful for authors to emphasize some caveats and qualifications to the generality of their study. What might happen in the many lakes that are long and narrow? Do you expect multiple gyres? Could the authors comment on whether the Rossby number is time dependant in a real lakes, due to increases in buoyancy forcing as the length of the day increases? This might mean change in sense of circulation patterns over the end of winter . How typical is the strong stratification that is as warm as 4C at bottom? Many large but relatively shallow lakes are less stratified and maybe be 2C or 3C at base (see https://aslopubs.onlinelibrary.wiley.com/doi/abs/10.1002/lno.11543 as one example). In these less stratified lakes you are much more likely to see gravity currents going to base of lake, so are circulation patterns are possibly different?*

**AR (Authors' response) 1**: The objective of this work is to provide a general characterization of the respective contribution of lake bathymetry and rotation on the warming of lakes under the ice. But the reviewer raises an important point regarding the basin's shape and the background stratification.

The study sites, Kilpisjärvi and Pavilion lakes, that we included in the discussion section, and where gyres were detected in the field, cannot be described as bowl-shape lakes. Pavilion Lake for example is an elongated lake and a gyre was measured at one of the regions where the lake widens. As far as we know, AUV transects were only conducted in this region of the lake (Forrest et al., 2013), so the presence of other gyres in other regions in this lake remains unknown. We think that not only the shape (surface and perimetral geometry), but the bathymetry itself will define the number of gyres that form (e.g, Akitomo et al., 2004). Especially relevant would be the presence of constrictions and/or shallow areas not only close to the lake shore but in the lake interior creating different "sub-basins". The modeling work by Huttula et al (2010) in Lake Pääjärvi suggests that indeed multiple gyres could form in more complex bathymetries. The role of the bathymetry in gyre formation would be definitely worth exploring in future work.

As the reviewer suggests, the effective buoyancy flux is expected to increase as the radiative forcing increases. In our numerical experiments, the effective buoyancy flux decays in time. In the field, however, the variations in solar radiation might lead to phases where the effective buoyancy flux is actually increasing in time, before reaching a point at which it starts decreasing due to the minimum in thermal expansivity. Thus, potentially yes, a lake initially in the geostrophic regime could move to the ageostrophic regime in the course of the Winter II period.

There are examples in the literature of lakes reaching temperatures close to $T_{MD}$ (or even higher) near the bottom (e.g., Bengtsson and Svensson,1996 (doi:10.2166/nh.1996.0018); Malm et al., (1998) (doi: 10.4319/lo.1998.43.7.1669); Cortés and McIntyre, 2019 (doi: 10.1002/lno.11296); Forrest et al., 2013). But, as the reviewer pointed out, many other lakes reach temperatures of only 2-3 degrees at the bottom of the lake. Certainly, the strength of the initial stratification (dρ/dz) at the start of the Winter II period (e.g., Jang et al., 2020) would affect the rate of deepening of the CML. For a given radiative flux and forcing time, a weaker background stratification would lead to deeper CMLs. The CML could potentially reach the base of the lake before the ice melts and then gravity currents would also flow downslope until the base of the lake. We do not see, though, why, there should be a change in the circulation pattern if the rotation regime remains the same.

We propose to expand the discussion section where we present the limitation of our approach with existing site-specific studies. We propose that the new text in the Discussion section reads: "*The numerical experiments described herein intend to provide a general characterization of the respective contribution of lake bathymetry and rotation on the warming of lakes under the ice; however, site-specific conditions will determine the actual response of a given lake. For example, although basin or sub-basin scale gyre formation has been reported to occur in lakes with bathymetries departing from the bowl-shaped one used in this study (e.g., Forrest et al, 2013; Kirillin et al., 2015), bathymetric effects could prevent gyre formation, or, by contrast, lead to the development of more complicated patterns (for example, multiple horizontal gyres, as in Huttula et al. (2010)). Also, the initial conditions could vary among lakes (e.g., Yang et al. 2020a). Although there are examples in the literature of lakes reaching values close to $T_{MD}$ near the lake bottom (e.g., Bengtsson and Svensson, 1996; Cortés and McIntyre, 2020; Forrest et al., 2013; Malm et al., 1998), other lakes reach temperatures of only 2-3 degrees at the bottom of the lake (e.g., Bouffard et al., 2016; Yang et al., 2020b). The radiative forcing conditions could also depart from those in this study, where a spatially-uniform radiative flux (e.g., Malm et al., 1997) or a steady daily radiative cycle (e.g., Bouffard et al., 2016) have been prescribed. The initial and forcing conditions (magnitude and time evolution) will influence the deepening rate of the CML, and/or the strength of differential heating and density currents, and will also determine the magnitude of Ro.*"

Akitomo, K., Kurogi, M. and Kumagai, M.: Numerical study of a thermally induced gyre system in Lake Biwa, Limnology, 5(2), 103–114, doi:10.1007/s10201-004-0122-9, 2004.

Bengtsson, L. and Svensson, T.: Thermal regime of ice covered Swedish lakes, Nord. Hydrol., 27(1–2), 39–56, doi:10.2166/nh.1996.0018, 1996.

Bouffard, D., Zdorovennov, R. E., Zdorovennova, G. E., Pasche, N., Wüest, A. and Terzhevik, A. Y.: Ice-covered Lake Onega: effects of radiation on convection and internal waves, Hydrobiologia, 780(1), 21–36, doi:10.1007/s10750-016-2915-3, 2016.

Malm, J., Terzhevik, A., Bengtsson, L., Boyarinov, P., Glinsky, A., Palshin, N. and Petrov, M.: Temperature and salt content regimes in three shallow ice-covered lakes: 1. Temperature, salt content, and density structure, Nord. Hydrol., 28(2), 99–128, doi:10.2166/nh.1997.0007, 1997.

Malm, J., Bengtsson, L., Terzhevik, A., Boyarinov, P., Glinsky, A., Palshin, N. and Petrov, M.: Field study on currents in a shallow, ice-covered lake, Limnol. Oceanogr., 43(7), 1669–1679, doi:10.4319/lo.1998.43.7.1669, 1998.

Yang, B., Wells, M. G., Li, J. and Young, J.: Mixing, stratification, and plankton under lake-ice during winter in a large lake: Implications for spring dissolved oxygen levels, Limnol. Oceanogr., lno.11543, doi:10.1002/lno.11543, 2020a.

Yang, B., Wells, M. G., McMeans, B. C., Dugan, H. A., Rusak, J. A., Weyhenmeyer, G. A., Brentrup, J. A., Hrycik, A. R., Laas, A., Pilla, R. M., Austin, J. A., Blanchfield, P. J., Carey, C. C., Guzzo, M. M., Lottig, N. R., Mackay, M. D., Middel, T. A., Pierson, D. C., Wang, J. and Young, J. D.: A New Thermal Categorization of Ice-covered Lakes, Geophys. Res. Lett., doi:10.1029/2020GL091374, 2020b.

*2) I was also wondering if the title might be slightly qualified? Rather than "Latitude an geometry", I'd suggest order be "geometry and latitude", as I think geometry is more important. You find the vast majority of dimictic lakes from about 40 to 70oN (Northern America and Northern Europe) where f varies from 0.935 to 1.367 10-4sˆ-1. So most of the variation in Rossby number between lakes is not primarily due to latitude, but rather their scale (and possibly magnitude of radiation which is indirectly also related to latitude).*

**AR2**: The reviewer is right. *Ro* is changing primarily due to *L*. We will modify the order of the words in the title as suggested: *"Bathymetry and latitude modify lake warming under the ice"*. However, we believe that the term bathymetry is more appropriate than geometry. The term geometry could be interpreted by the reader only in terms of lake dimensions and shape; however, *L* depends also on the width and depth of the shallow region, and so, on the topography of a given lake.

*3) I think you need to qualify that statement on line 20 that "Yet, as we move from planetary to smaller-scale systems, the importance of rotation in affecting convective processes remains overlooked. This is the case in lakes." I think you want to change or qualify the word "overlooked". In terms of convection there have many studies of the "spring thermal bar" where heating of lakes that are below 4oC drives a radial geostrophic flow near the shore, (https://en.wikipedia.org/wiki/Thermal_bar). This seems to be very closely related physics, so It would be worth reminding the reader of the connection between your study and this well known process in larger lakes (which is probably at low Ro end of your simulations). This is a well known example of Coriolis limiting heat transfer from edges to interior.*

*What I am confused about is that all observations and models thermal bar suggest you'd get an anti-clockwise flow (cyclonic) near the shore in Northern Hemisphere. I think this is opposite however to what is shown in low Rossby number case for Figure 3. There are many theoretical papers from 1970s on thermal bar, one example is https://doi.org/10.1080/03091927208236071 Huang, Joseph Chi Kan. "The thermal bar."ˇaGeophysical Fluid Dynamicsˇa3, no. 1 (1972): 1-25. This article mentions that "The results show a dominant meridional cyclonic flow along the perimetric edge of the lake and an anticyclonic flow in the middle portion of the lake. ˇa" I am confused what is the key difference between that classic field observation and your simulations for low Rossby number? I may be confused here, but if the sense of circulation is fundamentally different it would be useful to explain what is the key difference in setting up your simulations.*

**AR3**: The reviewer is correct. However, we believe that if we present the thermal bar in the introduction, this could lead the reader to think that this is a process covered in this study. Instead, in the introduction we propose to tone down our statement. We will change the text: *"Yet, as we move from planetary to smaller-scale systems, the importance of rotation in affecting convective processes **has drawn, comparatively, less attention**. This is the case in lakes".*

If the littoral region heats above $T_{MD}$ while the lake interior remains below $T_{MD}$ a thermal bar would form separating the two regions. The radial circulation in this case would be different to that in our ageostrophic regime. In our cross-section, we would see two thermal bars and four recirculating cells in the radial direction (one per littoral region and two at the lake interior). The water from both littoral regions, which is lighter than the water in the thermal bar ($\sim T_{MD}$), will move close to the surface towards the thermal bar and a return flow would form near the bottom. This is opposite to the radial circulation in the differential heating case under ice, where littoral waters are denser than pelagic waters. In the lake interior, water is also lighter than the water at the thermal bar region, so there would be a flow from the lake center towards the thermal bar near the surface and a return flow at deeper depths.

An opposite radial circulation would explain a different pattern in the azimuthal circulation. As the effect of rotation intensifies the water moving from the littoral region towards the thermal bar will be deflected at the surface towards the right, leading to the formation of a cyclonic gyre. In the lake interior, however, the flow at the surface is from the lake center towards the thermal bar, so when deflected to the right, it would lead to the formation of an anticyclonic gyre (e.g., Huang, 1972). This is opposite to the pattern described in this study. The pressure in our simulations is decreasing towards the littoral region, so, in the geostrophic regime an anticyclonic gyre develops in the lake.

We will include some lines in the discussion section. We propose that the text now reads:
"*To our knowledge, this circulation has not been reported in the field at times of under-ice radiatively-driven convection. The lake-wide anticyclonic circulation in Fig. 5b would be consistent with the inferred lake-wide anticyclonic gyre reported by Rizk et al (2014) at a time when circulation in lake Pääjärvi was dominated by a lateral gradient in the heat flux from the sediment and Ro was $O(10^{-3}\text{-}10^{-2})$ [...].* **Due to a different distribution of the pressure field (decreasing towards the littoral region in the under-ice differential heating case), the sense of the azimuthal circulation in Figs. 3g,h and Fig. 3k,l is, for example, opposite to that observed inshore of thermal bars (ice-free period), where a cyclonic gyre develops (e.g., Huang et al., 1972; Malm et al., 1993)**".

*Joseph Chi Kan Huang (1972) The thermal bar, Geophysical Fluid Dynamics, 3:1, 1-25, DOI: 10.1080/03091927208236071*

Malm, J., Grahn, L., Mironov, D. and Terzhevik, A.: Field investigation of the thermal bar in Lake Ladoga, spring 1991, Nord. Hydrol., 24(5), 339–358, doi:10.2166/nh.1993.12, 1993.

*4) There is a recent paper by Jazi Davarpanah et al. (2020) on rotating gravity currents using the Coriolis facility in Grenoble that goes into great detail on Rossby number effects and is a better reference than Wells, 2009 on line 32*
*https://agupubs.onlinelibrary.wiley.com/doi/abs/10.1029/2019JC015284*
*This study would also qualify your statement on line 149-150 that "When R/U > f gravity currents are affected by Earths rotation" - rather the large Grenoble experiments found that there is a gradual transition in gravity current dynamics that starts at Rossby number greater than 1,*

**AR4**: Thanks. We will include this reference on both lines.

*5) A number of studies in last 40 years have studied convection in rapidly rotating "dishes" - as an analog to understanding zonal jets in gas giants like Jupiter. Physically one might expect that these should have same or similar circulation patterns as your low Rossby number simulations (although they lack stratification). Hence it would be worth briefly commenting to what degree the circulation*

*patterns look similar or different to your Figure 3. I appreciate the experiments below are not stratified, but for many physicists these would be the rotating experiments they are familiar with. One recent example from Grenoble os*

*Read, P.L., Jacoby, T.N.L., Rogberg, P.H.T., Wordsworth, R.D., Yamazaki, Y.H., Miki-Yamazaki, K., Young, R.M., Sommeria, J., Didelle, H. and Viboud, S., 2015. An experimental study of multiple zonal jet formation in rotating, thermally driven convective flows on a topographic beta-plane.¢aPhysics of Fluids,¢a27(8), p.085111.*

*Before computer simulations were easier this type of system was also used in some high profile papers in 1980-1990s, see figure 2 in*

*Condie, Scott A., and Peter B. Rhines. "A convective model for the zonal jets in the atmospheres of Jupiter and Saturn."¢aNature¢a367, no. 6465 (1994): 711-713.*

*Sommeria, J., Meyers, S.D. and Swinney, H.L., 1989. Laboratory model of a planetary eastward jet. ˘aNature,¢a337(6202), pp.58-61.*

*also used to think about the modes of convection driven circulation under sloping geometry of Lake Vostok - see for instance the change from differential heating to columnar vortices in https://agupubs.onlinelibrary.wiley.com/doi/full/10.1029/2007GL032162*

**AR5**: We thank the reviewer for sharing these references. In our discussion section we already compared our circulation with the work by Fultz et al. (1959), which was conducted in a rotating cylinder and aimed at explaining the atmospheric large-scale circulation. We discussed there about the circulation patterns that we see in our simulations being analogue to those described by Fultz et al. (1959). In the discussion about the geostrophic regime, we specified that Fultz's work is conducted in laboratory rotating tanks ("The anticyclonic gyre circulation is also consistent with the Rossby wave regime reported in laboratory studies (rotating cylinder and annulus) mimicking the mid-to-high-latitude atmospheric circulation (Fultz et al., 1959)."). We will clarify also in the ageostrophic regime that this is a lab-scale work in the text and we will add other references.

In the discussion about the ageostrophic regime, we propose that the text now reads "*When Ro $\gtrsim 10^{-1}$, the horizontal heat transport is then accomplished by the ageostrophic components of the flow (downslope gravity currents). This cross-shore circulation might be considered analogue to the atmospheric Hadley cells,* **as reproduced in laboratory rotating-tank experiments (e.g., Fultz et al., 1959)**".

In the discussion about the geostrophic regime, we propose that the text reads "*The basin-scale anticyclonic gyre circulation is consistent with the Rossby wave regime reported in laboratory studies (rotating cylinder and annulus) mimicking the mid-to-high-latitude atmospheric circulation (e.g., Fultz et al., 1959; Sommeria et al., 1989 Condie and Rhines, 1994). The sense of rotation of the gyres in Figs. 3g,h and Fig. 3k,l is also consistent with these studies. Water temperature in the rotating tanks is above $T_{MD}$ and the sense of the gyre rotation is anticyclonic when the heating and cooling sources are provided at the center and the tank rim, respectively. The sense of rotation is reversed (cyclonic gyre) when the heating and cooling sources are exchanged. Within the Rossby regime, vortices and waves traveling cyclonically develop (Fig. 3h) and as Ro decreases, the wave lengths decrease and the gyre circulation is concentrated into jets that meander in the radial direction and could finally break (Condie and Rhines, 1994; Smith et al., 2014; Read et al., 2015). The presence of waves and/or vortices as in the scenario with Ro O($10^{-2}$) (Fig. 3h) is typical of transitional regimes (Fultz et al., 1959) and when they develop, the center of the anticyclonic gyre is not static in time but fluctuates laterally (video S1).*"

*6) There are a few more key studies on ice covered lakes that can be compared directly to the simulations. In particular in old studies on Tub lake, the scale and geometry looks like about exactly*

*scale as in the present student study. The lake is symmetric and has a profile from 0-4C, so is probably as similar as you could find, so a good question is whether the sense of circulation in studies by Likens is the same? They inferred basal heating was very important (as have other under ice studies during winter I.) but I feel this should be somewhat similar to radial differences in temperature gradients in winter II.*

*LIKENS, G. E., AND A. D. HASLER. 1962. Movements of radiosodium (NaZ4) within an ice-covered lake. Limnol. Oceansgr. 7: 48-56.*

*LIKENS, G. E., AND R. A. WACOTZKIE. 1965. Vertical water motions in a small icecovered lake. B. Geophys. Res. 70: 2333-2344.*

*Likens, G.E. and Ragotzkie, R.A., 1966. Rotary circulation of water in an ice-covered lake: With 6 figures and 1 table in the text. Internationale Vereinigung für theoretische und angewandte Limnologie: Verhandlungen, 16(1), pp.126-133.*

*Another old paper shows possible sinking near boundaries, consistent with your observations Welch, HE, & Bergmann, MA (1985). Water circulation in small arctic lakes in winter. Canadian Journal of Fisheries and Aquatic Sciences , 42 (3), 506-520.*

**AR6**: Thanks for sharing these key references. Indeed, by injecting radioactive tracers near the center and in the littoral region in Tub lake (45°N) in January 1962, Likens and Ragotzkie (1966) were able to detect the presence of a double-gyre circulation in the lake, with a cyclonic gyre in the lake interior (their Fig. 2), surrounded by an anticyclonic gyre in the littoral region (their Fig. 3). The azimuthal circulation pattern seems to be recurrent during the ice-on season, since Likens and Hasler (1962) already had detected the interior cyclonic circulation in January 1960.

They reported horizontal velocities of 30-35 m/day ($3.5 \times 10^{-4}$-$4 \times 10^{-4}$ m/s). For $L \sim 50$ m (the radius of bowl-shape Tub Lake), and $f = 1.028 \times 10^{-4}\,s^{-1}$ (45°N), the Rossby number, $Ro$, is $\sim 0.07$-$0.08$. This range lies within the values of Ro in our simulation in the ageostrophic regime and is consistent with the predicted circulation pattern. As the reviewer highlighted, in this case, and due to the snow cover on the lake, the circulation in the lake was thought to be driven by the input of heat from the sediment (Winter I).

Welch and Bergmann (1985) certainly identified the presence of density currents by adding dye (rhodamine) near the bottom of the littoral region ($\sim$ 4-m depth) in Methane Lake (63°N) during the Winter I period. They estimated through fluorometry that the approximate advance of the density current was $\sim$10 m/day (=$1 \times 10^{-4}$ m/s). However, if we use this velocity value, and for $L \sim 100$ m and $f = 1.3 \times 10^{-4}\,s^{-1}$ (63°N), the Rossby number should be $\sim$0.009 and the system should be in the transitional or geostrophic regimes. The authors reported that they "*were unable to detect the dye laterally more than a few meters on either side of a line between the addition site and the center of the lake*", suggesting no sign of the presence of a gyre in the lake. This mismatch between the expected and observed regime would suggest that other processes could play a role during Winter I period. Another possibility for the mismatch, is the uncertainty in their reported velocity estimates. The authors through the manuscript recurrently refer to "approximate" distances or "approximate limits of detectability". Given that lake horizontal currents are of O($10^{-4}$-$10^{-3}$) m/s during the Winter I period, there is probably a non-negligible error associated with the method used. Both Likens and Ragotzkie (1966) and Welch and Bergmann (1985) used indirect methods to estimate current velocities. This range of velocities were already a limitation for later studies using direct measurements. For example, Malm et al. (1998) (doi: 10.4319/lo.1998.43.7.1669) used an acoustic current meter to directly measure horizontal velocities under ice and they reported a threshold limit for the device of 0.2 mm/s, resolution of 0.2 mm/s and ± 40% accuracy of reading.

In any case, we will include both studies in the subsection "*Conceptual model for lake circulation*" in the Discussion section. We propose that the text there now reads: "*A double-gyre circulation was also reproduced in numerical simulations (Huttula et al., 2010) of early winter conditions in Lake Pääjärvi (61° N), when under-ice circulation was dominated by the input of heat*

*from the sediment instead of by radiatively-driven convection. Also, when the input of heat from the sediment dominated lake circulation, Likens and Ragotzkie (1966) injected radioactive tracers near the center and in the littoral region in Tub Lake (45° N, R~50 m and $A_{total}$ = 8.4 × 10$^{-3}$ km$^2$) and detected the presence of a double-gyre circulation when Ro ~0.1 (calculated with their indirect estimates of horizontal velocities of 3.5 × 10$^{-4}$-4 × 10$^{-4}$ m s$^{-1}$). The central cyclonic circulation had already been detected in this same lake by Likens and Hasler (1962) in a previous winter, suggesting that this azimuthal circulation pattern is recurrent during the ice-on season in the lake* ". In the discussion of the geostrophic regime, we propose that the text now reads: "*The lake-wide anticyclonic circulation in Fig. 5b would be consistent with the inferred lake-wide anticyclonic gyre reported by Rizk et al (2014) at a time when circulation in lake Pääjärvi was dominated by a lateral gradient in the heat flux from the sediment (Winter I period) and Ro was O(10$^{-3}$-10$^{-2}$). Nonetheless, Welch and Bergman (1985), reported radial velocities of 1 x 10$^{-4}$ m s$^{-1}$ during the Winter I period in Methane Lake (63°N, R ~100 m, and $A_{total}$ < 0.1 km$^2$) that would lead to estimates for Ro of O(10$^{-3}$-10$^{-2}$). By adding a dye (rhodamine) in a point in the littoral region and close to the lake bed, they detected the presence of density currents flowing offshore and no sign of gyre formation. This would be contrary to the expected radiatively-driven lake circulation in the geostrophic regime as presented in this study, and suggests that (1) other processes could be at play during Winter I or that (2) the radial velocity magnitude, and thus Ro, was underestimated by the authors. The latter is possible given that Welch and Bergman (1985) used dye concentrations to indirectly estimate O (10$^{-4}$) m s$^{-1}$ radial velocities in the lake.*"

*I hope all these comments are helpful in providing some more context to your interesting simulations.*

We are grateful for this very helpful review.

---

## Author Comment (AC2) · 7 Jan 2021

**Response to Referee #2**

*By application of a circulation model in an idealized domain mimicking the heating of an ice-covered lake of irregular morphometry by solar radiation, the authors arrive at an insightful demonstration of the rotation effects on the radial density flows produced by differential heating between shallow and deep lake areas. Rotational gravity flows are widespread in geophysical fluids and an advance in their quantification makes a valuable contribution to earth and planetary fluid dynamics. The ice-covered lakes represent rare natural examples, where these flows can be observed and investigated in detail at their whole range of scales, undisturbed by more energetic flows, usually persisting in open water, oceans, or the atmosphere. In that sense, the authors discuss an intriguing problem, of interest for a wide research community. The modeling methods are relevant, and the results are presented in a well-structured way.*

We would like to thank Referee #2 for his/her insightful review. It has greatly helped us identify points that needed better clarification.

*I had the opportunity to read the previous comment and generally share the concerns of the Reviewer1: my major criticism refers to the weak connection of the model configuration to the real conditions met in lakes and, as a result, misleading, superfluous, and over-generalized conclusions made by the authors.*
*Instead of nondimensionalization of the problem with regard to the rotational forces prior applying a numerical model, the authors voluntary choose the domain dimensions of $O(10^2)$ m and vary the Coriolis parameter within 2 (!) orders of magnitude. It is left for the reader's inspiration to imagine where on Earth $f = O(10^{-2})$ $s^{-1}$ can be observed (Line 139, Table 1). By using a posteriori re-scaling based on the Rossby number (Eq. 4, Line 148), a conclusion can be drawn that the ageostrophic regime (Ro = $O(10^{-1})$, Fig. 3, first column), similar to that described by Ulloa et al. (2019), can be found only in small ponds with an area of several ha. In lakes with characteristic length scales of $O(1)$ km (Ro = $O(10^{-2})$, Fig. 3, second column) and longer (Ro = $O(10^{-3})$, Fig. 3, second column), the shallow near-shore areas are effectively decoupled from the lake interior by rotation. The modeling results do not however provide a final proof for the importance of differential heating even in small ponds: they are typically much shallower than the modeled domain and have the background mixing intensities higher than those adopted in the model (Lines 108-111).*
*Herewith, the following outcomes of the study must be made clear: 1. For the vast majority of ice-covered lakes, differential heating does not contribute to the vertical mixing in the lake interior. 2. The previous findings of Ulloa et al. (2019) must be reconsidered taking into account the new results. 3. All variations of the Rossby number should be clearly related to corresponding variations in lake horizontal dimensions. Any mentioning of latitudinal effects should be removed, since for all seasonally ice-covered lakes $f = O(10^{-4})$ $s^{-1}$.*

**AR (Authors' response) 1**: We appreciate the reviewer's comment. However, some of the interpretations of the reviewer are not aligned with what we actually show throughout the manuscript. We believe that this reflects that our manuscript needs clarifications, which will be addressed.

- First: Starting already in the introduction we presented the Rossby number as the non-dimensional parameter to evaluate "*the importance of Earth rotation on horizontal flows*". So this non-dimensional number defines the rotational regime. The selection of the different simulations in Table 1 is based on this parameter. In our simulations we certainly modify $f$ and not the size of the lake, but this decision is based on computational efficiency in terms of the needed computational resources. Modifying $f$ instead of $L$ is, for example, common practise in laboratory (e.g., Afanasyev and Zhang, 2018; Cenedese and Adduce, 2008 (doi: 10.1017/S0022112008001237);Fultz et al., 1959; Wells and Cossu, 2013 (doi: 10.1098/rsta.2012.0366) and modeling (e.g., Carpenter and Timmermans, 2014 (10.1175/JPO-D-13-098.1); Pal and Chalamalla, 2020 (10.1017/jfm.2020.94); Ulloa et al., 2015 (doi: 10.1017/jfm.2015.311)) works studying rotational effects . We never stated in the text that a value of $f$ = O($10^{-2}$-$10^{-3}$) s$^{-1}$ is representative of Coriolis values on Earth, which obviously is not. We are sorry if our writing leads to that interpretation. What is representative of lakes on Earth is the order of magnitude of the Rossby number (as shown in Fig. 5d) and this was stated when we defined the scenarios "*We investigate three scenarios ranging from weak (Ro O($10^{-1}$)) to stronger (Ro O($10^{-3}$)) rotational influence (Table 1 and see Sect. 2.4 and Sect. S1.1 in the supplementary material for the Ro calculations). This range of Ro spans the expected range of values typical of the varying size and latitudinal distribution of ice-covered lakes on Earth (see Sect. 4).*" We believe that the source of confusion comes from the order in which we presented the information in the Methods section. We will combine the last paragraph in section 2.3 in Methods with the section 2.4 (*"Rossby number"*) to help clarify this point. We will also remove the column of $f$ values in Table 1.

- Second: As shown in Fig. 5d (calculated with the data of lakes in the HydroLAKES database) and discussed in the text, we expect the ageostrophic regime to be more common than the geostrophic regime. This suggests that for the vast majority of ice-covered lakes, the circulation studied by Ulloa et al. (2019) applies and differential heating could potentially affect the warming rates in the lake interior. Considering radial velocities in the range of O($10^{-3}$-$10^{-2}$) m/s as reported for ice-covered lakes, to obtain values of $Ro$ in the range 0.1 < $Ro$ < 1, the length $L$ from the littoral region to the lake interior in the $Ro$ calculations could be up to several kilometers even at high latitudes (see figure below). We will clarify this point in the text. We propose to include this clarification in the subsection "Rossby number" in the Methods section.

[Figure]

Figure. Example of range of lengths (here lake radius) and latitudes for the ageostrophic regime ($0.1 \lesssim Ro < 1$) for radial velocities of 0.005 m/s (black) and 0.05 m/s (red)

- The objective of this work is to provide a general framework of the importance that the lake bathymetry and rotation may have on the warming of lakes under the ice. Although we did not provide a specific "field site" validation, we validated the RANS model with a LES model in the supporting information and we provided examples in the discussion section that support the circulation pattern described here and which provide qualitative validation of this work.

Thus, based on these points, we have to disagree with the main two outcomes as expressed by the reviewer. Differential heating could indeed contribute to the warming of the lake interior. However, we agree with the reviewer on the fact that $Ro$ depends predominantly on the horizontal dimensions of the lake. While $L$ could vary by several orders of magnitude, $f$ remains $O(10^{-4})$ s$^{-1}$. Still, as shown in the figure above, the variability in $f$ alone will allow that bigger lakes lie within the ageostrophic regime as we move towards lower latitudes. See for example how a lake at 30°N could double the size of a lake at latitudes > 70°N and still be in the ageostrophic regime. The relative importance of $L$ and $f$ will be clarified in the text. As suggested by reviewer 1, we will also reorder the title of the manuscript.

We propose that Section 2.4 will be now called "Rossby number and test cases" and that the text in this section now reads: "*Since we are interested in evaluating the advection of heat from the littoral to the lake interior, the surface radius, R, was selected as the characteristic length scale in the calculations of Ro. Ro was calculated using the maximum radial velocity in the littoral region, $U_{rs-max}$, as the characteristic velocity scale (see details in Sect. S1.1 in supplementary material).*

$$Ro = \frac{U_{rs-max}}{fR}. \tag{4}$$

*$R/U_{rs-max}$ is the nominal time required for a gravity current at speed $U_{rs-max}$ to reach the center of the lake. When $R/U_{rs-max} > f^{-1}$, gravity currents are affected by Earth rotation (e.g., Davarpanah Jazi et al., 2020). While R could vary several orders of magnitude among the ice-covered lakes on Earth, f remains $O(10^{-4})$ s$^{-1}$. Thus, Ro depends predominantly on the horizontal dimensions of the lake. Still, the variability in f alone allows that bigger lakes lie within the same rotation regime as one moves towards lower latitudes. Note that for a given value of $U_{rs-max}$, to obtain the same value of Ro, Eq. (4) shows that*

*a lake at 30° N (f = 0.7×10⁻⁴ s⁻¹) would still double the size of a lake at latitudes ≥ 70° N (f ≥ 1.4×10⁻⁴ s⁻¹)*

$^1$$)$

*The model was first used to simulate rotational effects as in Ulloa et al. (2019), with a characteristic Ro O(10⁻¹). This corresponds to run 1 in Table1. For a range of measured radial velocities of O(10⁻³–10⁻²) m s⁻¹ under ice (Forrest et al., 2013; Kirillin et al., 2015; Rizk et al., 2014), a value of Ro O(10⁻¹) could be representative of lakes ranging from several tens of meters to several kms in length.  To analyze the effect of rotation in the lake circulation and in the warming of the CML, two additional simulations were conducted where we increased rotational effects by decreasing Ro up to two orders of magnitude (runs 2 and 3 in Table 1). To analyze bathymetric effects (differential heating), an additional simulation was conducted (reference simulation) where forcing was kept as in run 1, but the bathymetry was modified to obtain a cylinder of depth D = H. Each run spans 12 radiative cycles (12 days). This number of cycles was long enough to expose and analyze the effect of rotation and bathymetry on lake warming under ice.*"

.

*Other remarks:*

*L68 The geometrical factor G (Eq. 1 and Eq. 8) is of little predictive power as long as the hypsometry (the shape of the basin) is not included. When derived in a strict way, G incorporates a "shape factor" S = 0..1, which is found as an integral $S = \int_0^1 D(x)dx$, where D(x) = 0..1 is dimensionless depth, x = 0..1 is the relative distance from the shore to the lake center. For vertical walls S = 0, for linear slope S = 0.5, for the typical "bowl"-shaped lake S ≈ 0.3, and for the authors' tanh-approximation S ≈ 0:6. Hence, application of uncorrected G to different basin shapes can lead to ≥ 2 times differences in the result. Removal of G and related discussion is strongly recommended unless the basin shape is incorporated in the scaling.*

**AR2**: We would like to highlight that the *G* parameter is accounting for the advective transport of heat from the littoral region within the CML. Therefore, it is not the "basin-scale" hypsometry that is affecting *G*, but the "fraction" shallower than the depth of the CML. We agree with the reviewer on the importance of the lake hypsometry, and, lake hypsometry is directly included in the formulation of the *G* parameter. The geometrical factor *G* in Eq. 1 is expressed as

$$G = \left| \frac{A_{shallow}}{A_{total}} \left( \frac{\bar{h}}{h_{cml}} - 1 \right) \right|$$

For a given lake bathymetry, we calculate $A_{shallow}$ as the surface area of the water columns of the lake that are shallower than the depth of the convective mixed layer, and as shown in Fig. S1, the average depth of the littoral region $\bar{h}$ is calculated as

$$\bar{h} = \frac{V_{shallow}}{A_{shallow}}$$

where $V_{shallow}$ is the volume of the littoral region with an arbitrary morphology. So $\bar{h}$ would be an average value for the whole littoral region, not for a specific cross-section. Then, $A_{shallow}$, $V_{shallow}$ and $\bar{h}$ depend on the lake hypsometry. For the extreme case of vertical walls, $S = 1$ and, in Eq. 1, $A_{shallow}$ would be equal to zero, so $G = 0$. In Eq. 8, this G parameter was expressed for the specific case of a circular surface area, but again $L_{shallow}$ and $\bar{h}$ are average values for the whole basin.

*L131: 1/λ = 2.5 m⁻¹ (< 1 m Secchi depth) is rather turbid than moderately clear and is not typical for the majority of ice-covered lakes. Would the differential heating increase in more transparent waters? How the transparency affects the rotation effects? Make it clear in the text.*

**AR3**: Yes, the reviewer is right. A lake with a light attenuation of 2.5 m$^{-1}$ is better classified as turbid. We will modify this in the text.

There are different scenarios with respect to the attenuation of solar radiation: (1) The penetration depth of solar radiation is shallower than the depth of the littoral region and $h_{cml}$; (2) the penetration depth is deeper than the depth of the littoral region but shallower than $h_{cml}$; and (3) the penetration depth is deeper than $h_{cml}$. In the absence of horizontal advection of heat, the vertically-integrated rate of change of temperature in a water column of depth $d$ in the littoral region would be

$$\frac{\partial \overline{T_L}}{\partial t} \approx \frac{I_0\left(1-e^{-d/\lambda}\right)}{d}$$

And in the lake interior (considering that the background stratification suppresses the vertical transport of heat at $h_{cml}$)

$$\frac{\partial \overline{T_I}}{\partial t} \approx \frac{I_0\left(1-e^{-h_{cml}/\lambda}\right)}{h_{cml}}$$

Thus, the subtraction of the two gives the rate of change of the temperature between the two regions, that is:

$$\frac{\partial(\overline{T_L}-\overline{T_I})}{\partial t} \approx \frac{I_0\left(1-e^{-d/\lambda}\right)}{d} - \frac{I_0\left(1-e^{-h_{cml}/\lambda}\right)}{h_{cml}}$$

In the first scenario, $\lambda \ll d$; thus, $e^{-d/\lambda} \approx e^{-hcml/\lambda} \approx 0$ for all possible values of $\lambda$ and the lateral temperature gradient does not depend on the light attenuation: $\frac{\partial(\overline{T_L}-\overline{T_I})}{\partial t} \approx I_0(\frac{1}{d} - \frac{1}{h_{cml}})$ . In this scenario, more vigorous convection is expected as attenuation increases, but differential heating should not be affected. In the second scenario, if by decreasing the attenuation an important fraction of the solar radiation reaches the sediment of the littoral region and is absorbed there without being emitted back into the lake (no sediment heat flux), then $e^{-d/\lambda}$ may not be negligible and $\frac{\partial(\overline{T_L}-\overline{T_I})}{\partial t} \approx I_0(\frac{1-e^{-d/\lambda}}{d} - \frac{1}{h_{cml}})$. For a given radiative flux, differential heating would be weakened as the water becomes clearer. If otherwise we consider that all the incoming heat is retained in the water column, differential heating would not be modified. Finally, in the third scenario, $e^{-hcml/\lambda}$ may not be negligible, so if $(1-e^{-d/\lambda})\sim1$ (perfect insulator), differential heating would be enhanced as the water becomes clearer: $\frac{\partial(\overline{T_L}-\overline{T_I})}{\partial t} \approx I_0(\frac{1}{d} - \frac{1-e^{-h_{cml}/\lambda}}{h_{cml}})$.

Attenuation could then influence the magnitude of the radial velocity (and thus *Ro*), but we expect this effect to be secondary to the effect of the magnitude of $I_0$ and the geometry of the littoral region. We will write some lines about this in the Method section. We propose that the text in section 2.4 now reads *"Here the time t is expressed in days, $I_0$ (= $1\times10^{-5}$ ºC m s$^{-1}$) is the water surface radiative forcing, F(t) = sin(2πt) during the day (t < 0.5) and zero otherwise, and 1/λ (= 2.5 m$^{-1}$) is the depth scale for light attenuation. The order of magnitude of $I_0$ and the value for λ selected, are representative of late-winter conditions in turbid waters (Leppäranta et al., 2003; Bouffard et al., 2019; Ulloa et al., 2019).* **The intensity of convection is expected to decrease as λ increases (e.g., Winters et al., 2019), but the effect of light attenuation on differential heating and on the magnitude of the radial velocities remains secondary compared to the effect of the magnitude of $I_0$ and the geometry of the littoral region.** *For visualization purposes only, our results are shifted 0.25 days so the peak in the radiative heat flux matches midday (Fig. 1)."*

Leppäranta, M., Reinart, A., Erm, A., Arst, H., Hussainov, M. and Sipelgas, L.: Investigation of ice and water properties and under-ice light fields in fresh and Brackish water bodies, Nord. Hydrol., 34(3), 245–266, doi:10.2166/nh.2003.0006, 2003.

Winters, K. B., Ulloa, H. N., Wüest, A. and Bouffard, D.: Energetics of radiatively-heated ice-covered lakes, Geophys. Res. Lett., 2019GL084182, doi:10.1029/2019GL084182, 2019.

Although this is indeed an interesting topic, we consider that including the full explanation (the three scenarios described above) would deviate from the main point of the manuscript. In conclusion, we prefer not developing extensively this point in the main manuscript unless the referee and the handling editor think otherwise.

*L202-206: The geostrophic balance does not hold true in the bottom boundary layer (BBL), and the Coriolis effect on bottom-slope currents is strongly reduced by bottom friction. How good is BBL is reproduced by the model?*

**AR4**: We set no-slip conditions on the bottom boundaries. Also, in our simulations $h_{cml}$ reaches values of ~7 m after 12 days. The vertical resolution of our grid in this region is of 0.05 (up to a depth of 2 m) and 0.1 m (up to 10 m). For an expected BBL height of O(m), this means that our model resolves the BBL with at least 10 grid points.

*L314 Avoid the term "fjord-type" lakes, because the effect of the non-unity horizontal aspect ratio was not investigated in the study.*

**AR5**: We agree with the reviewer on the fact that using the term "fjord" will lead the reader to think not only on high slopes, but most probably, on an elongated lake basin. The text will be modified and will refer now only to lakes with "more vertical walls".

*L320 ". . .Peruvian Andes . . . ": Any example of a seasonally ice-covered lake at latitudes below 15°? A lake at 20° lat or lower can develop ice cover only at altitudes where liquid water is extremely rare. The whole discussion on the latitudinal variability is vague and should be rethought in terms of lake size (see above).*

**AR6**: Yes, we will restrict our discussion in the text and Fig. 5d to lakes starting at ~30°N (Lakes in the Himalayas and Tibet Plateau). Still, there are examples of lakes that seasonally freeze in the North of Chile, like Licancabur Lake (22°S) (Hock et al., 2002; bibcode 2002AGUFM.P71A0435H). With respect to the discussion about the importance of *L* vs. *f*, please see answer AR1.

*L321-323 The sentence is unsupported and—to be straight—wrong and misleading. Has to be removed.*

**AR7**: We believe that this sentence is not wrong or misleading and that the confusion is coming from the reviewer's interpretation that the ageostrophic regime only occurs in small shallow ponds (please, see details in answer AR1).

*L330-332 It is quite an interesting point deserving more discussion in view of the presented results. If the littoral water temperatures reach the maximum density point (TMD), but the lake interior stays colder than TMD, the ice cover will quickly melt over the shallows, forming the well-known "moats". As long as the rest of the lake stays ice covered, water temperatures in these open areas will be close to TMD without long-lasting stable stratification. "Moat" formation has been traditionally*

referred to terrestrial heat fluxes; the role of heat capture by rotation was never considered in this context but deserves a closer discussion.

**AR8**: Yes, as the reviewer points out, the excess heat retained in the littoral region could act as an internal accelerator for the ice melting process, and thus, potentially, for moat formation. Although this is indeed an interesting topic, we believe that an extended discussion linking the retention of heat in the shallows and moat formation would overstate the significance of the results of this numerical study. Instead, we propose that we add a new line in this section establishing a possible connection between the two processes. We propose that the text reads: "*Due to the retention of heat in the littoral region, water temperature there could potentially reach values $\geq T_{md}$. This would lead to the development of a stable stratification in the littoral region and to the suppression of convection that, in contrast, would continue in the lake interior. **This could** have implications for early formation of thermal bars and/or **contribute to the formation of moats (e.g., Nolan et al., 2002)**"*.

Presentation in form of a HESS publication of the otherwise well-performed and insightful study to a wider community can be recommended only after resolving the above mentioned issues.

---

## Editor Comment (EC1) · Bettina Schaefli (Editor) · 12 Jan 2021

This paper has received two very detailed reviews and both reviewers state in their formal recommendation that the scientific significance, the scientific quality and the presentation quality are good to excellent but ask for a second review round before publication. I therefore invite the authors to submit a revised version, incorporating the modifications explained in the public discussion. These modifications mostly relate to additional or restructured explanations of the used methods and additional references to extend the discussion of the results.

---

## Referee Report (RR1)

The authors have adjusted the study, removing the most obvious discrepancies between the modeling results and their interpretation with regard to the real physical objects (ice-covered lakes). The remaining arguments on the role of the latitudinal variability of the Coriolis force are not fully convincing, but they are put in a more or less acceptable context. The study can be published with a couple of minor but important (!) corrections, allowing the dedicated reader to get an unbiased picture of the presented results:

- Line 155: "...to several kilometers in length" should be replaced with "up to the maximum of $\approx 1$ km".

  Explanation: the order-of-magnitude analysis, when performed properly, should use the fixed $f = \mathcal{O}(10^{-4})$. The variations of $\pm 30$ % from the mean value are neglected in the order-of-magnitude estimations. Even if accounting for the minor variations of $f$, in the given velocity range of two orders of magnitude and $Ro = \mathcal{O}(10^{-1})$, the maximum length scale is $\sim 1.6$ km, which is far below "several kilometers". As the study clearly shows, the ageostrophic, $Ro = \mathcal{O}(10^{-1})$, regime is generally *not applicable* to lakes with the horizontal scales $> 1$ km.

- Line 352: After "...should be common under ice ..." a phrase has to be added: "for lakes with horizontal dimensions $\lesssim 1$ km"

  Explanation: see the remarks above. Here, the over-generalization (should be common under ice ...) is not supported and can misguide the reader, because $Ro = \mathcal{O}(10^{-1})$ is only true for small ice-covered lakes with relatively high horizontal velocities (irregular hypsography).

- Line 298: "Winter I period". It is better to remove the word "period". Depending on climatic conditions, a lake may have only "Winter I", or only "Winter II" throughout the entire ice-covered period. The terms "Winter I/II" have not became widespread yet; it is better to cite Kirillin et al. (2012) at their first appearance to provide the reader with a necessary background.

---

## Author Response (AR2)

**Authors' response**

**Response to Referee #1**
Submitted on 22 Jan 2021

*Accepted as is*

We thank again Referee #1 for his/her suggestions to improve this manuscript.

**Response to Referee #2**
Submitted on 05 Feb 2021

*The authors have adjusted the study, removing the most obvious discrepancies between the modeling results and their interpretation with regard to the real physical objects (ice-covered lakes). The remaining arguments on the role of the latitudinal variability of the Coriolis force are not fully convincing, but they are put in a more or less acceptable context. The study can be published with a couple of minor but important (!) corrections, allowing the dedicated reader to get an unbiased picture of the presented results:*

We would like to thank again Referee #2 for his/her thorough review of this manuscript

*Line 155: \. . . to several kilometers in length" should be replaced with \up to the maximum of ≈ 1 km".*
*Explanation: the order-of-magnitude analysis, when performed properly, should use the fixed f = O($10^{-4}$). The variations of ±30 % from the mean value are neglected in the order-of-magnitude estimations. Even if accounting for the minor variations of f, in the given velocity range of two orders of magnitude and Ro = O($10^{-1}$), the maximum length scale is ~ 1.6 km, which is far below "several kilometers". As the study clearly shows, the ageostrophic, Ro = O($10^{-1}$), regime is generally not applicable to lakes with the horizontal scales > 1 km.*

**AR (Authors's response) 1:** We have clarified this in the text. Even if we use $f = 10^{-4}$ s$^{-1}$, the ageostrophic regime would approximately cover $0.1 \lesssim Ro \lesssim 1$ (note that this work does not define the exact limit between the ageostrophic and transition regimes). If we use $Ro = 0.1$ (the lower approximate limit of this regime), $L$ would be 0.5 km for $U = 0.005$ m s$^{-1}$, but 5 km for $U = 0.05$ m s$^{-1}$. $L$ represents one-half of the width of the lake (the radius if of circular shape) so, lakes with lengths up to approximately 10 km could potentially still be in this regime. We will clarify that we refer to the total length ($2L$) in the text. But, we agree with the reviewer that the sentence "lakes ranging from several tens of meters to several kilometers in lengths" may lead to confusion and now we express this length also in terms of order of magnitude, so the text now reads "*For a range of measured radial velocities of O($10^{-3}$–$10^{-2}$) m s$^{-1}$ under ice (Forrest et al., 2013; Kirillin et al., 2015; Rizk et al., 2014), **a value of Ro O($10^{-1}$) could be representative of lakes of O ($10^{-1}$-$10^{0}$) km in length (~2L)**"*

.

*Line 352: After ". . . should be common under ice . . . " a phrase has to be added: "for lakes with horizontal dimensions ≲ 1 km".*

*Explanation: see the remarks above. Here, the over-generalization (should be common under ice . . . ) is not supported and can misguide the reader, because Ro = O($10^{-1}$) is only true for small ice-covered lakes with relatively high horizontal velocities (irregular hypsography).*

**AR2**: This sentence refers to Fig. 5b, where we see that this regime would be common at all latitudes. We have clarified in the text that this is because there is a bias in the lake size distribution towards smaller lakes. More than 85% of lakes on Earth have surface area < 1 km$^2$ (Messager et al., 2016). The text now reads "***The bias in the distribution of lake size towards smaller lakes (more than 85% of lakes on Earth have surface areas smaller than 1 km$^2$, Messager et al., 2016)*** *and the bias in the potential for ice cover formation (e.g. Sharma et al., 2019) and the distribution of lakes on Earth towards mid to high latitudes (> 45° N, Fig. 5b) indicates that the ageostrophic regime with Ro O($10^{-1}$) should be common under ice as shown by the mean (red lines) values in Fig. 5d.*"

*Line 298: "Winter I period". It is better to remove the word "period". Depending on climatic conditions, a lake may have only "Winter I", or only "Winter II" throughout the entire ice-covered period. The terms "Winter I/II" have not became widespread yet; it is better to cite Kirillin et al. (2012) at their first appearance to provide the reader with a necessary background.*

**AR3**: We agree with the reviewer. We have removed the term "period" after "Winter I" and "Winter II" and have included the reference to Kirillin et al. (2012), the first time we mentioned them in the text.